# RE-PO: Robust Enhanced Policy Optimization as a General Framework for LLM Alignment

**Xiaoyang Cao**[1*], **Zelai Xu**[2*], **Mo Guang**[3], **Kaiwen Long**[3], **Michiel A. Bakker**[1],
**Yu Wang**[2], **Chao Yu**[4†]

[1]IDSS, Massachusetts Institute of Technology, [2]EE, Tsinghua University, [3]Li Auto Inc.,
[4]SIGS, Tsinghua University

## Abstract

Standard human preference-based alignment methods, such as Reinforcement Learning from Human Feedback (RLHF), are a cornerstone technology for aligning Large Language Models (LLMs) with human values. However, these methods are all underpinned by a strong assumption that the collected preference data is clean and that all observed labels are equally reliable. In reality, large-scale preference datasets contain substantial label noise due to annotator errors, inconsistent instructions, varying expertise, and even adversarial or low-effort feedback. This creates a discrepancy between the recorded data and the ground-truth preferences, which can misguide the model and degrade its performance. To address this challenge, we introduce **R**obust **E**nhanced **P**olicy **O**ptimization (**RE-PO**). RE-PO employs an Expectation-Maximization algorithm to infer the posterior probability of each label's correctness, which is used to adaptively re-weigh each data point in the training loss to mitigate noise. We further generalize this approach by linking a broad class of preference losses to induced probabilistic models. This enables systematic robustification of existing alignment algorithms while preserving exact probabilistic equivalence for likelihood-style losses. Theoretically, under perfect calibration and a population/full-batch setting, we show that RE-PO recovers the true annotator reliability. Our experiments demonstrate RE-PO's effectiveness as a general framework, generally enhancing four state-of-the-art alignment algorithms (DPO, IPO, SimPO, and CPO) against their corresponding standard versions. When applied to Mistral and Llama 3 models, the RE-PO-enhanced methods improve AlpacaEval 2 win rates by up to 7.0% over their respective baselines. Our code and other resources are available at repo-alignment.github.io.

## 1 Introduction

Aligning Large Language Models (LLMs) with human values is a critical prerequisite for developing safe and reliable AI systems. Reinforcement Learning from Human Feedback (RLHF) has emerged as the dominant paradigm for this task (Christiano et al., 2017; Ziegler et al., 2019; Ouyang et al., 2022). To mitigate the complexity and instability of the traditional RLHF pipeline, simpler and more direct methods such as Direct Preference Optimization (DPO) (Rafailov et al., 2023) have been developed, which reframe alignment as a classification-like problem.

However, these alignment methods implicitly assume that preference datasets provide a clean and reliable approximation of a single ground-truth preference signal. In practice, this assumption is often violated. Large-scale preference datasets are typically aggregated from multiple crowdworkers or teacher models, and are therefore subject to substantial label noise arising from inattention, misunderstanding, or systematic bias (Frénay & Verleysen, 2013; Gao et al., 2024). Empirical analyses suggest that a significant fraction (often between 20% and 40%) of preference pairs in modern alignment datasets may be corrupted or inconsistent (Gao et al., 2024). Classic work on learning with noisy labels shows that standard loss functions can overfit such corrupted supervision and suffer

---

*Equal contribution: `xycao@mit.edu`, `zelai.eecs@gmail.com`.
†Corresponding authors: `yuchao@sz.tsinghua.edu.cn`.

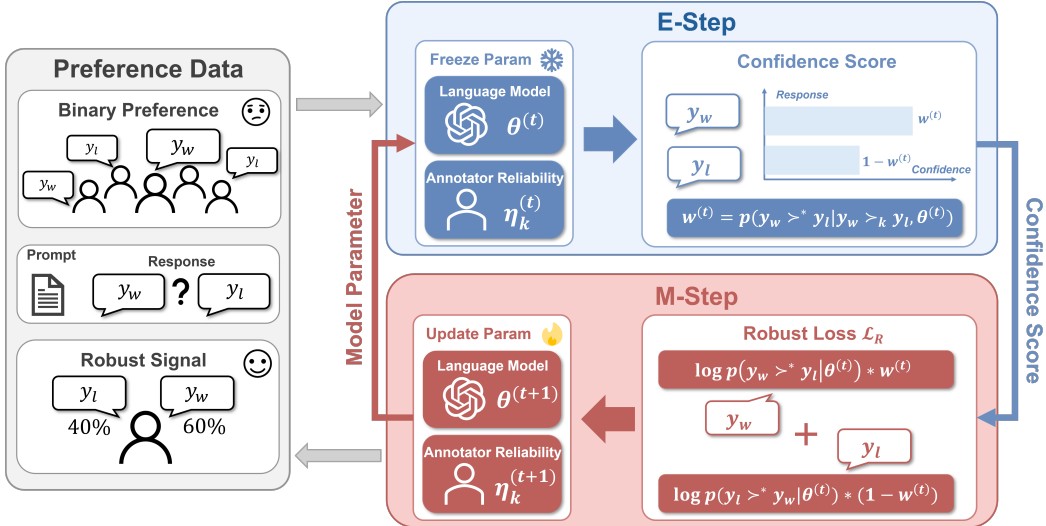

Figure 1: Overview of the Robust Enhanced Policy Optimization (RE-PO) framework. Starting from noisy pairwise feedback, RE-PO uses an Expectation-Maximization (EM) procedure to jointly refine label confidences and the policy. In each iteration, the E-step estimates a confidence score for every observed preference by inferring the posterior probability that the label is correct under the current model and annotator reliabilities. The M-step then uses these scores as adaptive weights to update both the LLM policy and the annotator reliability parameters, progressively down-weighting likely corrupted labels and emphasizing reliable supervision.

severe degradation in generalization performance (Natarajan et al., 2013). Similar observations are reported in the label-noise survey of Frénay & Verleysen (2013). In the context of LLM alignment, Gao et al. (2024) further demonstrate that even a 10 percentage point increase in the label-noise rate can lead to drops of tens of percentage points in downstream win rates, highlighting the practical importance of robustness to noisy preference data.

To address this challenge, we propose Robust Enhanced Policy Optimization (RE-PO). Instead of assuming that every observed label is a fixed ground truth, our approach aims to learn a preference model that remains accurate and stable even when the training data contains substantial noise. The core innovation of RE-PO is its departure from the hard labels used in traditional RLHF. Rather than committing to binary supervision, we treat the correctness of each observed preference as a latent variable and compute soft confidence weights over labels, so that highly reliable feedback contributes more strongly while suspicious pairs are down-weighted. Building on Expectation-Maximization-style approaches to learning from unreliable annotators in crowdsourcing (Dawid & Skene, 1979; Chen et al., 2013), RE-PO employs an Expectation-Maximization (EM) framework that simultaneously models annotator reliability while optimizing the LLM. In the E-step, it infers the posterior probability that each annotated label is correct, effectively estimating annotator reliability. In the M-step, it uses these probabilities as adaptive weights to update the LLM, thereby learning from a dynamically re-weighted preference signal.

Our experiments validate RE-PO as an effective general framework. We show that applying RE-PO generally enhances four state-of-the-art alignment algorithms (DPO, IPO, SimPO, and CPO) across two different base models (Mistral-7B and Llama-3-8B) on the AlpacaEval 2 benchmark (Table 2). In our main results, RE-PO-enhanced methods achieve substantial win-rate gains on AlpacaEval 2, with improvements of up to 7.0 percentage points over their standard counterparts. Furthermore, we theoretically analyze when RE-PO can recover annotator reliability (Theorem 4.1) and empirically verify this behavior in controlled experiments (Section 5.5).

In summary, our contributions are as follows:

- We propose Robust Enhanced Policy Optimization (RE-PO), a principled EM-based algorithm that treats the correctness of each preference label as a latent variable, jointly infers per-label

(and per-annotator) reliabilities, and uses them as adaptive weights in the training loss, yielding LLM alignment that is substantially more robust to noisy and inconsistent feedback.

- We theoretically establish a generalized RE-PO framework by using the Gibbs construction to connect a broad class of preference losses to induced probabilistic models. This lifts RE-PO from a single algorithm to a general framework, enabling standard methods such as DPO, IPO, SimPO, and CPO to be systematically transformed into robust counterparts; for non-likelihood losses, the induced objective can differ from the original one.

- We conduct extensive experiments demonstrating the practical effectiveness and versatility of RE-PO. Across four alignment algorithms, two base models (Mistral-7B and Llama-3-8B), and AlpacaEval 2, RE-PO delivers consistent win-rate improvements of up to 7.0 percentage points, and further shows clear gains on a real multi-annotator dataset (MultiPref), along with qualitative and visual analyses of how it down-weights low-confidence, noisy labels.

## 2 RELATED WORK

**LLM alignment with hard preference labels.** The standard paradigm for aligning Large Language models (LLMs) with human values is Reinforcement Learning from Human Feedback (RLHF), which involves training a reward model and then fine-tuning the policy against it (Christiano et al., 2017; Ouyang et al., 2022). To mitigate the complexity and instability of this multi-stage process, a family of simpler, direct alignment algorithms has emerged (Rafailov et al., 2023; Azar et al., 2023; Meng et al., 2024; Hong et al., 2024). These methods bypass the explicit reward modeling stage by optimizing a direct classification-style loss on the preference data. However, a critical limitation shared by these methods is their reliance on hard preference labels. This approach models human feedback as a definitive, binary choice, treating every label with equal and absolute confidence. Consequently, it is highly vulnerable to the significant label noise present in real-world datasets, as standard loss functions can lead models to overfit to corrupted labels (Natarajan et al., 2013; Zhang & Sabuncu, 2018; Frénay & Verleysen, 2013). A simple annotation error, such as an accidental misclick, is given the same weight as a deliberate, high-quality judgment. This inability to distinguish between reliable feedback and noise means that the model's performance degrades significantly as the error rate increases (Frénay & Verleysen, 2013; Gao et al., 2024). In contrast, soft-label approaches that represent preferences probabilistically can better accommodate uncertainty in feedback by assigning confidence scores or weights to individual labels (Müller et al., 2019; Song et al., 2024). By allowing the learning algorithm to rely on high-quality signals while down-weighting noise, such approaches provide a natural path toward robust preference alignment. This is precisely the perspective adopted by our RE-PO framework, which replaces hard labels with EM-estimated soft confidences.

**Learning from noisy feedback.** The vulnerability to label noise situates preference alignment within the classic machine learning problem of Learning with Noisy Labels (LNL) (Natarajan et al., 2013; Frénay & Verleysen, 2013). Foundational work in this area, such as the Dawid–Skene model (Dawid & Skene, 1979), uses an EM algorithm to simultaneously infer true latent labels while estimating annotator reliability. This principle was later extended to pairwise comparisons in the Crowd-BT model (Chen et al., 2013), which jointly estimates item scores and annotator-specific reliability parameters in crowdsourced ranking tasks. In modern LLM alignment, several methods have been proposed to improve robustness to noisy preference data. These can be broadly divided into loss-centric approaches and data-centric filtering strategies. In the first category, ROPO (Liang et al., 2024) proposes an iterative robust preference optimization procedure that jointly applies a noise-tolerant loss and down-weights (or discards) highly uncertain samples, without relying on external teacher models. rDPO (Chowdhury et al., 2024) constructs an unbiased estimator of the true loss but requires the global noise rate to be known a priori. Hölder-DPO (Fujisawa et al., 2025) introduces a loss with a "redescending" property, which inherently nullifies the influence of extreme outliers without needing a known noise rate. In the second category, Selective DPO (Gao et al., 2025) proposes filtering examples based on their difficulty relative to the model's capacity—a concept orthogonal to label correctness—using validation loss as a proxy.

Our proposed RE-PO framework is complementary to these methods. Rather than only modifying the loss shape or discarding high-loss points, RE-PO explicitly models the data-generating process by treating annotator reliability and label correctness as latent variables to be inferred. This allows

Table 1: Formulations of the preference loss ($\mathcal{L}_{\text{pref}}$) for prominent alignment algorithms.

| Method | Preference Loss $\mathcal{L}_{\text{pref}}(x, y_w \succ y_l)$ |
|---|---|
| DPO (Rafailov et al., 2023) | $-\log \sigma \left( \beta \log \frac{\pi_\theta(y_w\|x)}{\pi_{\text{ref}}(y_w\|x)} - \beta \log \frac{\pi_\theta(y_l\|x)}{\pi_{\text{ref}}(y_l\|x)} \right)$ |
| IPO (Azar et al., 2023) | $\left( \log \frac{\pi_\theta(y_w\|x)}{\pi_{\text{ref}}(y_w\|x)} - \log \frac{\pi_\theta(y_l\|x)}{\pi_{\text{ref}}(y_l\|x)} - \frac{1}{2\beta} \right)^2$ |
| SimPO (Meng et al., 2024) | $-\log \sigma \left( \frac{\beta}{\|y_w\|} \log \pi_\theta(y_w\|x) - \frac{\beta}{\|y_l\|} \log \pi_\theta(y_l\|x) - \gamma \right)$ |
| CPO (Xu et al., 2024) | $-\log \sigma (\beta \log \pi_\theta(y_w\|x) - \beta \log \pi_\theta(y_l\|x)) - \log \pi_\theta(y_w\|x)$ |

RE-PO to assign fine-grained, example-specific weights based on a posterior confidence, providing a principled way to separate signal from noise.

## 3 METHODOLOGY

This section details our proposed RE-PO algorithm. We first review the standard DPO framework in Section 3.1. In Section 3.2, we introduce a latent-variable model that explicitly distinguishes clean and corrupted preference labels. Section 3.3 then derives the corresponding EM-based update rules for RE-PO. Section 3.4 presents a practical mini-batch implementation for RE-PO.

### 3.1 PRELIMINARIES: DIRECT PREFERENCE OPTIMIZATION

The goal of preference alignment is to fine-tune a language model policy, $\pi_\theta$, using a dataset of preferences $\mathcal{D} = \{(x, y_w, y_l)_i\}_{i=1}^N$, where response $y_w$ is preferred over $y_l$ for a given prompt $x$. Direct Preference Optimization (DPO) (Rafailov et al., 2023) offers a simple and effective method for this, bypassing the complex multi-stage pipeline of traditional RLHF (Christiano et al., 2017; Ouyang et al., 2022). DPO directly optimizes the policy by minimizing a simple classification loss:

$$\mathcal{L}_{\text{DPO}}(\pi_\theta, \pi_{\text{ref}}) = -\mathbb{E}_{(x, y_w, y_l) \sim \mathcal{D}} \left[ \log \sigma \left( \beta \log \frac{\pi_\theta(y_w|x)}{\pi_{\text{ref}}(y_w|x)} - \beta \log \frac{\pi_\theta(y_l|x)}{\pi_{\text{ref}}(y_l|x)} \right) \right], \quad (1)$$

where $\sigma(\cdot)$ is the sigmoid function, $\pi_{\text{ref}}$ is a fixed reference policy and $\beta$ is a scaling hyperparameter.

### 3.2 RE-PO FRAMEWORK: CORE ASSUMPTIONS

A critical limitation of DPO is its implicit assumption that all observed preferences in $\mathcal{D}$ are correct. In practice, this data is often noisy. To address this, we propose Robust Enhanced Policy Optimization (RE-PO), which is built upon two core assumptions that reframe the problem.

**Assumption 1: A Latent noise-free preference.** We assume that for each training example $(x_i, y_{w,i}, y_{l,i})$ there exists an underlying noise-free preference, denoted $y_{w,i} \succ^* y_{l,i}$, which represents the label we would obtain in the absence of annotation errors. The observed preference $y_{w,i} \succ_{k_i} y_{l,i}$ (provided by annotator $k_i$) is treated as a potentially corrupted observation of this ground truth. To model this, we introduce a binary latent variable $z_i \in \{0, 1\}$ for each data point, where $z_i = 1$ if the observed label matches the latent noise-free preference and $z_i = 0$ otherwise. The reliability of annotator $k$ is parameterized by $\eta_k \triangleq p(z_i = 1 \mid k_i = k)$. Here $k_i \in \{1, \ldots, K\}$ denotes the index of the annotator who provided the $i$-th label, and $K$ is the total number of annotators in the dataset.

**Assumption 2: A general probabilistic model for preferences.** Building on this latent variable model, we must also define the probability of the noise-free preference itself, $p(y_w \succ^* y_l | x, \theta)$. To accommodate preference losses beyond DPO (e.g., IPO (Azar et al., 2023)), we instantiate RE-PO with a broad class of loss functions, $\mathcal{L}_{\text{pref}}$. For likelihood-style losses, this recovers the original probabilistic interpretation; for non-likelihood losses, it should be viewed as an induced robust surrogate. Table 1 provides representative examples used in prominent alignment algorithms.

To connect these diverse loss functions to a unified probabilistic view, we draw inspiration from the Boltzmann distribution (Luce, 1959). We define an induced preference probability where each

ordering is scored by $\exp(-\mathcal{L}_{\text{pref}})$. This yields the following definition for the noise-free preference probability:

$$p(y_w \succ^* y_l | x, \theta) = \sigma\left(\mathcal{L}_{\text{pref}}(x, y_l \succ y_w; \theta) - \mathcal{L}_{\text{pref}}(x, y_w \succ y_l; \theta)\right), \tag{2}$$

where $\sigma(\cdot)$ is the sigmoid function. This construction induces a well-defined binary probability model for each loss. For instance, with the standard DPO loss, this equation recovers the Bradley-Terry model (Bradley & Terry, 1952) (see Sections A and B for derivations). For non-likelihood losses, this induced probability may differ from the exact original training objective.

## 3.3 THE RE-PO ALGORITHM VIA EXPECTATION-MAXIMIZATION

Based on these core assumptions, we aim to find the parameters $\theta$ and $\boldsymbol{\eta}$ that maximize the marginal log-likelihood of the observed data. The probability of a single observed preference is obtained by marginalizing over the latent variable $z_i$:

$$p(y_{w,i} \succ_{k_i} y_{l,i} | x_i, \theta, \boldsymbol{\eta}) = p(y_{w,i} \succ^* y_{l,i} | x_i, \theta)\eta_{k_i} + p(y_{l,i} \succ^* y_{w,i} | x_i, \theta)(1 - \eta_{k_i}). \tag{3}$$

Directly maximizing $\sum_i \log p(y_{w,i} \succ_{k_i} y_{l,i})$ is feasible but coupled and non-convex due to the log-sum form. We therefore employ the EM algorithm (see details in Section C), which provides stable alternating updates with closed-form reliability updates. In this iterative process, the superscript $(t)$ will denote the values of parameters at iteration $t$.

**E-Step: Inferring label correctness.** In the E-step, given the current parameters $\theta^{(t)}$ and $\boldsymbol{\eta}^{(t)}$, we compute the posterior probability $w_i$ that the $i$-th observed label is correct. This value $w_i$ acts as a "soft label" or the model's confidence in the data point.

$$w_i^{(t)} = \frac{p(y_{w,i} \succ^* y_{l,i} | x_i, \theta^{(t)})\eta_{k_i}^{(t)}}{p(y_{w,i} \succ^* y_{l,i} | x_i, \theta^{(t)})\eta_{k_i}^{(t)} + p(y_{l,i} \succ^* y_{w,i} | x_i, \theta^{(t)})(1 - \eta_{k_i}^{(t)})}. \tag{4}$$

where $p(y_{w,i} \succ^* y_{l,i} | x_i, \theta^{(t)})$ and $p(y_{l,i} \succ^* y_{w,i} | x_i, \theta^{(t)})$ can be computed according to equation 2.

**M-Step: weighted parameter update.** In the M-step, we update the policy parameters $\theta$ and reliabilities $\boldsymbol{\eta}$ using the confidences $w_i^{(t)}$ computed in the E-step. This step conveniently separates into two independent updates.

First, the policy is updated by minimizing a weighted loss function. As established in Assumption 2, our probabilistic model for $p(y_w \succ^* y_l)$ can be instantiated with multiple preference losses $\mathcal{L}_{\text{pref}}$, making RE-PO a practical meta-framework across common objectives. The general RE-PO loss is:

$$\mathcal{L}_{\text{RE-PO}}(\theta) = -\sum_{i=1}^{N} \left[ w_i^{(t)} \log p(y_{w,i} \succ^* y_{l,i} | x_i, \theta) + (1 - w_i^{(t)}) \log p(y_{l,i} \succ^* y_{w,i} | x_i, \theta) \right]. \tag{5}$$

Second, the reliability $\eta_k$ for each annotator is updated to the average confidence of all labels they provided. This has a simple and efficient closed-form solution:

$$\eta_k^{(t+1)} = \frac{\sum_{i \in \mathcal{I}_k} w_i^{(t)}}{N_k}. \tag{6}$$

Here we define the index set of labeled pairs as $\mathcal{I}_k = \{i : k_i = k\}$, and the number of labels as $N_k$.

## 3.4 PRACTICAL IMPLEMENTATION WITH MINI-BATCH TRAINING

While the exact M-step updates are clear, performing a full iteration over the entire dataset to re-calculate the annotator reliabilities $\boldsymbol{\eta}$ after each policy update step can be computationally expensive. To balance computational efficiency and performance, we introduce a more practical online update for $\eta_k$ using an Exponential Moving Average (EMA). Instead of a hard assignment, we perform a soft update based on the statistics from the current mini-batch $\mathcal{B}$:

$$\eta_k \leftarrow (1 - \alpha)\eta_k + \alpha \cdot \frac{\sum_{i \in \mathcal{B} \cap \mathcal{I}_k} w_i}{N_{k,\mathcal{B}}}. \tag{7}$$

Here, $N_{k,\mathcal{B}}$ is the number of examples from annotator $k$ in the current mini-batch, and $\alpha \in (0,1]$ is a momentum hyperparameter. The complete training procedure for RE-PO is summarized in Algorithm 1:

---

**Algorithm 1:** Mini-batch Implementation of Robust Enhanced Policy Optimization (RE-PO)

---

**Input:** Dataset $\mathcal{D} = \{(x_i, y_{w,i}, y_{l,i}, k_i)\}_{i=1}^N$; Base policy $\pi_\theta$, reference policy $\pi_{\text{ref}}$; Preference loss $\mathcal{L}_{\text{pref}}$; Hyperparameters: learning rate $\lambda$, epochs $E$, EMA momentum $\alpha$, initial annotator reliabilities $\eta_k \in [0.5, 1]$ for all $k \in \{1, \dots, K\}$

1 **for** *epoch = 1 to E* **do**
2     **for** *batch $\mathcal{B} \subset \mathcal{D}$* **do**
3        For each sample $i \in \mathcal{B}$, compute $w_i$ using current $\theta$ and $\eta_{k_i}$ via equation 4;
4        Compute the weighted loss $\mathcal{L}_{\text{RE-PO}}(\theta)$ for the batch via equation 5;
5        Update parameters $\theta$ using an optimizer (e.g., AdamW (Loshchilov & Hutter, 2019));
6        **for** *each annotator $k$ present in the batch* **do**
7           Update $\eta_k$ via equation 7;
8        **end**
9     **end**
10 **end**

---

## 4    THEORETICAL ANALYSIS OF RE-PO

The robustness of RE-PO stems from its adaptive weighting mechanism. This section first provides an intuitive analysis of these training dynamics and then formalizes this intuition with theoretical guarantees, demonstrating that the RE-PO framework can recover the true reliability of annotators.

At the start of training, when the language model is not yet well-optimized, its predictions are uncertain, and the probabilities $p(y_w \succ^* y_l | x, \theta)$ are close to 0.5. The confidence score $w_i$ approximates the annotator's reliability, $\eta_{k_i}$. The loss then acts as a form of label smoothing, preventing the model from being severely misled by incorrect labels early on. As the policy improves, its behavior adapts. For a high-quality label, the model predicts a high probability for the winning response, and $w_i$ approaches 1, causing the loss to function like a standard preference optimization objective. Conversely, $w_i$ approaches 0 for a noisy label. The loss is then dominated by the $(1 - w_i)$ term, which flips the optimization direction toward the true preference.

We now formalize the intuition that RE-PO can recover the true reliability of annotators. We provide this analysis under an idealized setting: full-batch training where the M-step for the policy parameters $\theta$ is assumed to have converged perfectly. While our practical implementation in Algorithm 1 uses mini-batch gradient updates (a form of Generalized EM), this idealized analysis provides a strong theoretical justification for our framework.

Consider the update rule in equation 6, defined as an operator $T_k(\eta)$. The following theorem is stated at the population/expectation level and shows that iterating this operator recovers the true annotator reliability under the idealized assumptions.

**Theorem 4.1** (Identification and convergence of RE-PO). *Let $\theta^\star$ be a perfectly calibrated parameter such that the model distribution matches the ground-truth preference distribution. Assume that under annotator-$k$'s data distribution, the model probabilities are not almost surely equal to $\frac{1}{2}$.*

*Consider the sequence of reliability estimates $\{\eta_k^{(t)}\}_{t \geq 0}$ generated by the population operator $\eta_k^{(t+1)} = T_k(\eta_k^{(t)})$. Then, for any initialization $\eta_k^{(0)} \in (0, 1)$, the iterates converge to the true reliability $\eta_k^\star \triangleq \mathbb{E}[z_i \mid k_i = k]$:*

$$\lim_{t \to \infty} \eta_k^{(t)} = \eta_k^\star.$$

The proof is provided in Appendix D. In section 5.5, we empirically corroborate that the practical finite-sample mini-batch procedure closely tracks this population-level behavior.

**Practical implications and limitations.** The assumption of a perfectly calibrated model in Theorem 4.1 is intentionally idealized: in practice, we apply RE-PO to base models that are not exactly calibrated to the ground-truth preference distribution. In our experiments, we always start from strong instruction-tuned LLMs (`Mistral-7B-Instruct-v0.2` and `Meta-Llama-3-8B-Instruct`), which already display good zero-shot preference behavior. Empirically, we do not observe the failure mode suggested by an extremely misaligned initialization:

across the broad range of hyperparameters explored in Section 5.4, the learned $\eta_k$'s remain stable and the downstream performance consistently improves over the corresponding base methods. Furthermore, the controlled experiments in Section 5.5, where we inject substantial synthetic noise into the data, show that RE-PO's estimated reliabilities closely track the ground-truth values, suggesting robustness to imperfect calibration in practice. If the base LLM were initialized in a highly misaligned regime, the E-step could assign misleadingly high confidence to incorrect labels and RE-PO might fail to effectively denoise the supervision.

## 5 EXPERIMENTS

In this section, we conduct a comprehensive set of experiments to evaluate the performance of RE-PO. We begin in section 5.1 by detailing our experimental setup, including the models, datasets, evaluation benchmarks, and baseline algorithms. In Section 5.2, we present our main results. Section 5.3 reports additional experiments to evaluate RE-PO's performance on realistic multi-annotator datasets. We then conduct an ablation study in Section 5.4 to analyze the framework's sensitivity to its key hyperparameters. In section 5.5, we provide an empirical verification of our theoretical claims of Theorem 4.1.

### 5.1 EXPERIMENTAL SETUP

**Models and training settings.** We use two state-of-the-art open-source large language models as our base models: `Mistral-7B-Instruct-v0.2` and `Meta-Llama-3-8B-Instruct`. For fine-tuning, we utilize two datasets from the SimPO paper (Meng et al., 2024), which were generated via on-policy sampling using prompts from the UltraFeedback dataset (Cui et al., 2024). The specific datasets are `mistral-instruct-ultrafeedback` for the Mistral model and `llama3-ultrafeedback-armorm` for the Llama-3 model.[1] As these datasets do not provide annotator-specific information, we model the preferences as if they originate from a single, virtual annotator ($K = 1$).[2] In addition to these UltraFeedback-based datasets, we further evaluate RE-PO on the real-world MultiPref multi-annotator preference dataset (Miranda et al., 2024), where per-annotator reliabilities can be explicitly modeled (Section 5.3).

**Evaluation benchmarks.** We assess model performance on two widely recognized evaluation benchmarks. The first is AlpacaEval 2 (Dubois et al., 2024), an automatic, LLM-based evaluator that measures model performance by computing the win rate against reference outputs. It provides both a raw Win Rate (WR) and a Length-Controlled (LC) Win Rate to account for verbosity bias. The second is Arena-Hard (Li et al., 2024), a challenging benchmark composed of difficult prompts crowdsourced from the LMSYS Chatbot Arena. It is designed to differentiate high-performing models by testing them on complex, real-world user queries. Performance is reported as the win rate against a suite of other models.

**Baseline algorithms.** To demonstrate that RE-PO operates as a versatile meta-framework, we benchmark it against four popular direct preference alignment methods: DPO (Rafailov et al., 2023); IPO (Azar et al., 2023), which uses a squared hinge loss to optimize preferences; SimPO (Meng et al., 2024), which proposes a simplified, reference-free reward formulation normalized by sequence length; and CPO (Xu et al., 2024), which adds a term to directly maximize the likelihood of the preferred response. For each of these baselines, we compare the original algorithm to its RE-PO-enhanced counterpart (e.g., DPO vs. RE-DPO). In addition, we include robustness-oriented baselines rDPO (Chowdhury et al., 2024) and Hölder-DPO (Fujisawa et al., 2025), as well as simple label-smoothing variants for each method. The results are shown in Table 2.

---

[1] See Appendix I for links to models and datasets.

[2] This is a reasonable simplification. For instance, a pool of two annotators with reliabilities $\eta_A$ and $\eta_B$, appearing with frequencies $p_A$ and $p_B$ respectively, can be modeled as a single annotator with an effective reliability $\eta_{\text{unified}} = p_A\eta_A + p_B\eta_B$.

Table 2: Performance comparison on AlpacaEval 2 for `Mistral-7B-Instruct-v0.2` and `Meta-Llama-3-8B-Instruct` fine-tuned on UltraFeedback-based preference datasets. Metrics reported are **LC** (Length-Controlled Win Rate) and **WR** (Raw Win Rate), both in percentage points. The table presents reference *Baselines* (bottom) alongside four algorithm families (DPO, IPO, SimPO, CPO). For each family, we compare the *Standard* implementation, the variant with Label Smoothing (*w/ LS*), and RE-PO (*w/ RE-PO*). **Bold** denotes the best result within each family for a given backbone.

| Method | Mistral-7B-Instruct | | | Llama-3-8B-Instruct | | |
|---|---|---|---|---|---|---|
| | Standard | w/ LS | w/ RE-PO | Standard | w/ LS | w/ RE-PO |
| DPO | 28.5 / 28.6 | 29.7 / 27.5 | **35.5 / 33.0** | 40.8 / 42.9 | 41.3 / 42.6 | **44.1 / 46.2** |
| IPO | 30.8 / 28.0 | 29.7 / 28.7 | **32.9 / 30.5** | 43.6 / 41.6 | 40.3 / 38.2 | **48.3 / 48.6** |
| SimPO | 28.3 / 29.7 | 26.5 / 27.1 | **30.4 / 32.9** | 44.5 / 37.1 | **48.1** / 38.7 | 46.9 / **39.4** |
| CPO | 26.3 / 26.4 | **28.5 / 28.8** | 27.6 / 27.8 | 35.9 / 40.3 | 35.3 / 34.8 | **40.1 / 43.8** |
| Base Model | 21.1 / 16.5 | | | 29.7 / 29.9 | | |
| rDPO | 28.1 / 29.1 | | | 37.3 / 35.4 | | |
| Hölder-DPO | 30.1 / 28.6 | | | 39.3 / 38.2 | | |

Table 3: Performance of DPO and RE-DPO on AlpacaEval 2 when trained on the MultiPref dataset (Miranda et al., 2024). Results are reported as LC / WR (%) for `Mistral-7B-Instruct-v0.2` and `Meta-Llama-3-8B-Instruct`.

| Method | Mistral-7B-Instruct | Llama-3-8B-Instruct |
|---|---|---|
| DPO | 28.8 / 26.4 | 36.7 / 39.3 |
| RE-DPO (Ours) | **31.8 / 28.8** | **41.1 / 44.4** |

## 5.2 MAIN RESULTS

As shown in Table 2, our experimental results provide strong evidence that RE-PO generally improves preference-based alignment across objectives, model scales, and datasets. Below we highlight the main empirical findings.

**RE-PO as a general framework.** A first observation is that RE-PO behaves as a generally effective plug-in robustness layer for a wide range of alignment losses. Across all four objective families (DPO, IPO, SimPO, CPO) and both backbones (Mistral-7B and Llama-3-8B), the RE-PO-enhanced variant generally matches or outperforms the corresponding standard implementation on AlpacaEval 2. For example, on Mistral-7B, RE-DPO improves LC / WR from $28.5/28.6$ to $35.5/33.0$ (a gain of $+7.0$ and $+4.4$ points, respectively), and on Llama-3-8B, RE-IPO improves LC / WR from $43.6/41.6$ to $48.3/48.6$ (a gain of $+4.7$ and $+7.0$ points, respectively). These trends indicate that RE-PO is an effective robustness layer for existing preference objectives.

**Comparison with label smoothing and robust baselines.** Table 2 also compares RE-PO to two natural robustness baselines: label smoothing applied to each preference loss and the recently proposed robust objectives rDPO (Chowdhury et al., 2024) and Hölder-DPO (Fujisawa et al., 2025). Label smoothing sometimes yields modest gains over the standard objective (e.g., SimPO w/ LS on Llama-3-8B improves LC from $44.5$ to $48.1$), but RE-PO is competitive and often achieves the best performance within each family and backbone. For instance, in the DPO family, RE-DPO outperforms both label smoothing and the specialized robust baselines: on Llama-3-8B, RE-DPO reaches $44.1/46.2$ on AlpacaEval 2, compared to $41.3/42.6$ for DPO w/ LS, $37.3/35.4$ for rDPO, and $39.3/38.2$ for Hölder-DPO. These results suggest that explicitly modeling noisy supervision via RE-PO is more effective than purely loss-level modifications or global noise-correction schemes.

**Qualitative analysis of noisy labels.** Beyond aggregate metrics, we also perform a qualitative analysis of the learned confidence scores. In Appendix F, we present case studies of preference pairs with very low posterior confidence $w_i$. RE-PO assigns low confidence to annotations that are off-task,

Table 4: Ablation study on the initial annotator reliability ($\eta_0$) and the EMA momentum ($\alpha$). Results are reported for RE-DPO on `Mistral-7B-Instruct-v0.2` trained on UltraFeedback-based data, evaluated on AlpacaEval 2 (LC / WR) and Arena-Hard (WR), all in percentage points. The best-performing settings used in our main experiments are highlighted.

| Metric | Initial $\eta_0$ | | | | EMA $\alpha$ | | | | |
|---|---|---|---|---|---|---|---|---|---|
| | 0.99 | **0.9 (Ours)** | 0.75 | 0.55 | 0.001 | 0.01 | **0.1 (Ours)** | 0.5 | 1.0 |
| AlpacaEval2 LC (%) | 30.9 | **35.5** | 31.1 | 31.4 | 30.9 | 30.1 | **35.5** | 33.4 | 31.1 |
| AlpacaEval2 WR (%) | 31.7 | **33.0** | 33.3 | 32.0 | 27.8 | 27.2 | **33.0** | 34.8 | 28.9 |
| Arena-Hard WR (%) | 12.3 | **14.7** | 12.4 | 11.8 | 12.9 | 13.6 | **14.7** | 14.0 | 12.8 |

inconsistent with the prompt, or at odds with a more plausible alternative response. Together with the quantitative gains in Tables 2, these examples illustrate that RE-PO not only improves benchmark performance but also identifies and down-weights noisy supervision at the example level.

## 5.3 Multi-annotator experiments on MultiPref

To further evaluate RE-PO under realistic multi-annotator disagreement, we conduct additional experiments on the MultiPref dataset (Miranda et al., 2024), a large-scale human preference dataset with genuine rater disagreement. The official training split contains **227 unique human annotators**. Unlike the UltraFeedback-based datasets used in our main experiments, MultiPref provides annotator identifiers, allowing us to instantiate an individual reliability parameter $\eta_k$ for each annotator and to update these parameters via our EM-style scheme.

We train vanilla DPO and our RE-DPO on MultiPref for both `Mistral-7B-Instruct-v0.2` and `Meta-Llama-3-8B-Instruct`, and evaluate the resulting models on AlpacaEval 2. As summarized in Table 3, RE-DPO consistently outperforms vanilla DPO under this multi-annotator setup: for Llama-3-8B, the AlpacaEval LC improves from 36.7 to 41.1 and WR from 39.3 to 44.4; for Mistral-7B, LC improves from 28.8 to 31.8 and WR from 26.4 to 28.8. These gains mirror the trends observed in our UltraFeedback experiments and show that RE-PO remains beneficial when trained on data with heterogeneous, potentially noisy annotators, rather than a single virtual annotator.

In Appendix E, we visualize the learned annotator reliabilities distributions on MultiPref. Experiment results indicate that RE-PO identifies a high-reliability majority and a nontrivial tail of downweighted annotators, and that this pattern is robust across different prior settings and backbones. Moreover, to probe the impact of the choice of automatic judge, we repeat the MultiPref evaluation using a different LLM evaluator; Appendix G reports these results and shows that the performance gains from RE-DPO are stable across judge models.

## 5.4 Ablation study

We conduct ablation studies to analyze the sensitivity of RE-PO to two key hyperparameters: the initial annotator reliability $\eta_0$, and the EMA momentum parameter $\alpha$. All experiments are performed using RE-DPO on the Mistral-7B-Instruct-v0.2 model. The results are summarized in Table 4.

**Effect of initial $\eta_0$.** The initial reliability $\eta_0$ sets the model's prior belief about the correctness of the labels in the dataset. As shown in Table 4, the model's performance is best when $\eta_0$ is set to 0.9, which was the value used in our main experiments. An overly optimistic initialization (e.g., $\eta_0 = 0.99$) can cause the model to trust noisy labels too strongly at the beginning of training, hindering the denoising process. Conversely, a pessimistic initialization (e.g., $\eta_0 = 0.55$) treats the data as highly unreliable from the outset, which can slow down the model's ability to learn the underlying noise-free preference. An initial value of 0.9 appears to strike the right balance, starting with a reasonable assumption of data quality.

**Effect of EMA parameter $\alpha$.** The EMA parameter $\alpha$ governs the update rate of the annotator reliability scores, balancing the influence of historical estimates against new information from the current mini-batch. Our experiments confirm that the optimal performance is achieved with $\alpha = 0.1$. The model shows considerable sensitivity to this parameter. A very small $\alpha$ (e.g., 0.001) makes the

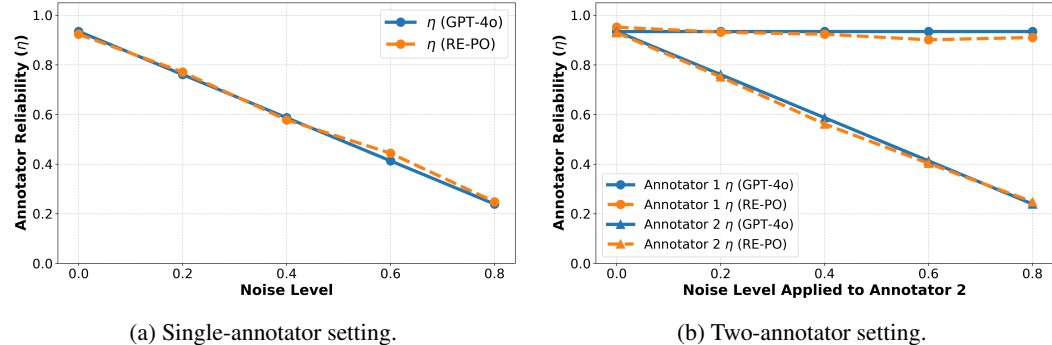

(a) Single-annotator setting.          (b) Two-annotator setting.

Figure 2: Empirical verification of annotator reliability estimation under controlled synthetic noise. Ground-truth reliability ($\eta$ GPT-4o) is established using GPT-4o's labels on UltraFeedback-derived preference pairs, and different reliability levels are simulated by injecting synthetic noise into copies of the dataset. In the single-annotator setting (a), a single annotator's dataset is perturbed with varying noise rates. In the two-annotator setting (b), Annotator 1 uses the original data with no added noise, while noise is progressively added to Annotator 2's data. The plots compare ground-truth reliabilities (solid lines) with RE-PO-estimated reliabilities (dashed lines), showing that RE-PO closely tracks the true reliability in both scenarios.

reliability updates exceedingly slow, preventing the estimates from adapting to the model's evolving understanding of the data. On the other hand, a very large $\alpha$ (e.g., 1.0) makes the updates highly volatile, as the reliability score becomes dependent solely on the samples in the current mini-batch.

## 5.5 EMPIRICAL VERIFICATION OF THEOREM 4.1

We conduct controlled experiments to verify Theorem 4.1. Our setup is designed to align with the theorem's assumption of a perfectly calibrated model, for which we use a small-scale base model, `Qwen2.5-0.5B-Instruct`, to ensure fast convergence. To simulate annotators with varying levels of reliability, we create distinct copies of the UltraFeedback dataset (Cui et al., 2024) for each annotator and inject a controlled degree of synthetic noise into their respective dataset.

We test two scenarios, with results presented in Figure 2: (a) **Single Annotator:** A single annotator whose dataset is modified with a synthetically controlled noise rate. (b) **Two Annotators:** A scenario with two annotators, where Annotator 1 serves as a baseline using the original data without added noise, while the dataset for Annotator 2 is injected with progressively increasing noise levels.

The results in Figure 2 show that the estimated reliability $\eta$ (RE-PO) closely tracks the ground-truth $\eta$ (GPT-4o) in both single-annotator (Figure 2a) and two-annotator (Figure 2b) settings. Notably, in the two-annotator experiment, RE-PO successfully identifies the stable reliability of the baseline annotator while accurately tracking the declining reliability of the noisy one. Although the theorem assumes a perfectly calibrated model, these experiments demonstrate that RE-PO's reliability estimates remain accurate and stable even when the underlying model is only approximately calibrated and trained under realistic noise patterns, mitigating concerns that early miscalibration would systematically down-weight correct labels.

## 6 CONCLUSION AND FUTURE WORK

In this paper, we introduce Robust Enhanced Policy Optimization (RE-PO), a framework designed to address the critical challenge of aligning LLMs with noisy human preference data. Our approach employs an Expectation-Maximization algorithm to infer the reliability of each preference pair, treating labels as soft, dynamic weights rather than fixed ground truths. Across the evaluated objectives and backbones, RE-PO improves multiple state-of-the-art alignment algorithms, achieving substantial gains over their corresponding standard versions. A natural limitation of our current theory is the assumption of a perfectly calibrated model; extending convergence guarantees to settings where the base model is significantly misaligned remains important future work.

## ACKNOWLEDGEMENTS

This research was supported by National Natural Science Foundation of China (No.62406159,62325405), Zhongguancun Project C20250301, Tsinghua University Initiative Scientific Research Program, TsinghuaEfort Joint Research Center for EAI Computation and Perception, Beijing National Research Center for Information Science, Technology (BNRist), Beijing Innovation Center for Future Chips, and State Key laboratory of Space Network and Communications.

## REPRODUCIBILITY STATEMENT

To facilitate reproducibility, we provide (i) full methodological details and derivations in Sections 3 and 4, with additional derivations and proofs in Appendix A–D; (ii) experimental setup, implementation details, and additional empirical results in Section 5 and the appendices (including runtime/resources in Appendix H and I); (iii) public artifacts, including the project webpage (repo-alignment.github.io) and the code repository (github.com/XiaoyangCao1113/RE-PO).

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

## A  DERIVATION OF GENERAL PROBABILISTIC MODEL

Here we provide the detailed derivation for equation 2. For a given prompt $x$ and candidate responses $y_w, y_l$, we assume the probability of the ground-truth preference $y_w \succ^* y_l$ is proportional to $\exp(-\mathcal{L}_{\text{pref}}(x, y_w \succ y_l))$. That is:

$$p(y_w \succ^* y_l | x, \theta) \propto \exp(-\mathcal{L}_{\text{pref}}(x, y_w \succ y_l)) \tag{8}$$

Similarly, for the inverse preference:

$$p(y_l \succ^* y_w | x, \theta) \propto \exp(-\mathcal{L}_{\text{pref}}(x, y_l \succ y_w)) \tag{9}$$

Since $y_w \succ^* y_l$ and $y_l \succ^* y_w$ are the only two mutually exclusive outcomes for a binary preference, their probabilities must sum to 1. Using the property of normalized probabilities from a proportional relationship, we have:

$$
\begin{aligned}
p(y_w \succ^* y_l | x, \theta) &= \frac{\exp(-\mathcal{L}_{\text{pref}}(x, y_w \succ y_l))}{\exp(-\mathcal{L}_{\text{pref}}(x, y_w \succ y_l)) + \exp(-\mathcal{L}_{\text{pref}}(x, y_l \succ y_w))} \\
&= \frac{1}{1 + \exp(-(\mathcal{L}_{\text{pref}}(x, y_l \succ y_w) - \mathcal{L}_{\text{pref}}(x, y_w \succ y_l)))} \\
&= \sigma\left(\mathcal{L}_{\text{pref}}(x, y_l \succ y_w) - \mathcal{L}_{\text{pref}}(x, y_w \succ y_l)\right)
\end{aligned}
$$

The last line is the General Probabilistic Model in equation 2.

## B    CONSISTENCY WITH BRADLEY-TERRY MODEL FOR DPO

We show that equation 2 is consistent with the Bradley-Terry model when applied to DPO. The DPO loss for a preferred pair $(y_w, y_l)$ given prompt $x$ is:

$$\mathcal{L}_{\text{DPO}}(x, y_w \succ y_l) = -\log \sigma \left( \beta \log \frac{\pi_\theta(y_w|x)}{\pi_{\text{ref}}(y_w|x)} - \beta \log \frac{\pi_\theta(y_l|x)}{\pi_{\text{ref}}(y_l|x)} \right) \tag{10}$$

Let $S(x, y_w, y_l) = \beta \log \frac{\pi_\theta(y_w|x)}{\pi_{\text{ref}}(y_w|x)} - \beta \log \frac{\pi_\theta(y_l|x)}{\pi_{\text{ref}}(y_l|x)}$. Then, we can write:

$$\mathcal{L}_{\text{DPO}}(x, y_w \succ y_l) = -\log \sigma(S(x, y_w, y_l))$$
$$\mathcal{L}_{\text{DPO}}(x, y_l \succ y_w) = -\log \sigma(S(x, y_l, y_w)) = -\log \sigma(-S(x, y_w, y_l))$$

Substituting these into our general probabilistic model (equation 2):

$$\begin{aligned}
p(y_w \succ^* y_l | x, \theta) &= \sigma \left( \mathcal{L}_{\text{DPO}}(x, y_l \succ y_w) - \mathcal{L}_{\text{DPO}}(x, y_w \succ y_l) \right) \\
&= \sigma \left( \log \sigma(S(x, y_w, y_l)) - \log \sigma(-S(x, y_w, y_l)) \right) \\
&= \sigma \left( \log \frac{\sigma(S(x, y_w, y_l))}{1 - \sigma(S(x, y_w, y_l))} \right) \\
&= \sigma(S(x, y_w, y_l)) \\
&= \sigma \left( \beta \log \frac{\pi_\theta(y_w|x)}{\pi_{\text{ref}}(y_w|x)} - \beta \log \frac{\pi_\theta(y_l|x)}{\pi_{\text{ref}}(y_l|x)} \right)
\end{aligned}$$

This resulting probability exactly matches the form of the Bradley-Terry model (Bradley & Terry, 1952) for preferences, where the implicit reward of a response $y$ is $r(x, y) = \beta \log \frac{\pi_\theta(y|x)}{\pi_{\text{ref}}(y|x)}$.

## C    DERIVATION OF THE RE-PO EM ALGORITHM

The primary objective of Robust Enhanced Policy Optimization (RE-PO) is to find the model parameters $\theta$ and the vector of annotator reliabilities $\boldsymbol{\eta}$ that maximize the log-likelihood of the observed data. The observed data consists of prompts, chosen and rejected responses, and the annotator's index, denoted as $X = \mathcal{D} = \{(x_i, y_{w,i}, y_{l,i}, k_i)\}_{i=1}^N$.

The log-likelihood function is given by:

$$\mathcal{L}(\theta, \boldsymbol{\eta}) = \sum_{i=1}^N \log \left[ p(y_{w,i} \succ^* y_{l,i} | x_i, \theta) \eta_{k_i} + p(y_{l,i} \succ^* y_{w,i} | x_i, \theta)(1 - \eta_{k_i}) \right] \tag{11}$$

There is a sum inside the logarithm, which makes direct optimization coupled and non-convex. The Expectation-Maximization (EM) algorithm is an iterative procedure designed to solve such maximum likelihood problems with latent variables by alternating between an Expectation (E) step and a Maximization (M) step.

### C.1    DERIVATION OF THE Q-FUNCTION (THE E-STEP)

The EM algorithm simplifies the problem by working with the complete data, $(X, Z)$, where $Z = \{z_i\}_{i=1}^N$ is the set of all latent variables.

The complete-data log-likelihood, $\mathcal{L}_c$, assumes that we know the values of all latent variables $z_i$:

$$\mathcal{L}_c(\theta, \boldsymbol{\eta}; X, Z) = \sum_{i=1}^N \left( z_i \log \left[ p(y_{w,i} \succ^* y_{l,i} | x_i, \theta) \eta_{k_i} \right] + (1 - z_i) \log \left[ p(y_{l,i} \succ^* y_{w,i} | x_i, \theta)(1 - \eta_{k_i}) \right] \right) \tag{12}$$

This form is tractable because the logarithm acts on products, which can be separated into sums.

The core idea of EM is to iteratively maximize the expectation of the complete-data log-likelihood. This expectation, known as the Q-function, is taken with respect to the posterior distribution of the latent variables $Z$, given the observed data $X$ and the parameter estimates from the current iteration, $(\theta^{(t)}, \boldsymbol{\eta}^{(t)})$.

$$Q(\theta, \boldsymbol{\eta}|\theta^{(t)}, \boldsymbol{\eta}^{(t)}) \equiv \mathbb{E}_{Z|X,\theta^{(t)},\boldsymbol{\eta}^{(t)}}[\mathcal{L}_c(\theta, \boldsymbol{\eta}; X, Z)] \tag{13}$$

To compute this expectation, we push the expectation operator inside the summation. The only random variables in $\mathcal{L}_c$ are the $z_i$.

$$Q(\theta, \boldsymbol{\eta}|\theta^{(t)}, \boldsymbol{\eta}^{(t)}) = \sum_{i=1}^{N} \left( \mathbb{E}[z_i] \log \left[ p(y_{w,i} \succ^* y_{l,i}|\theta)\eta_{k_i} \right] + (1 - \mathbb{E}[z_i]) \log \left[ p(y_{l,i} \succ^* y_{w,i}|\theta)(1 - \eta_{k_i}) \right] \right) \tag{14}$$

The term $\mathbb{E}[z_i]$ is the expectation of the binary variable $z_i$, which is its posterior probability of being 1. This probability is conditioned on the observed data and the parameters from the current iteration $t$. We denote this posterior probability as $w_i^{(t)}$, which is computed in the E-Step:

$$
\begin{aligned}
w_i^{(t)} &\equiv \mathbb{E}[z_i|X_i, \theta^{(t)}, \boldsymbol{\eta}^{(t)}] \\
&= p(z_i = 1|y_{w,i} \succ_{k_i} y_{l,i}, x_i, \theta^{(t)}, \boldsymbol{\eta}^{(t)}) \\
&= \frac{p(y_{w,i} \succ_{k_i} y_{l,i}|z_i = 1, x_i, \theta^{(t)})p(z_i = 1|k_i, \boldsymbol{\eta}^{(t)})}{p(y_{w,i} \succ_{k_i} y_{l,i}|x_i, \theta^{(t)}, \boldsymbol{\eta}^{(t)})} \\
&= \frac{p(y_{w,i} \succ^* y_{l,i}|x_i, \theta^{(t)})\eta_{k_i}^{(t)}}{p(y_{w,i} \succ^* y_{l,i}|x_i, \theta^{(t)})\eta_{k_i}^{(t)} + p(y_{l,i} \succ^* y_{w,i}|x_i, \theta^{(t)})(1 - \eta_{k_i}^{(t)})}
\end{aligned} \tag{15}
$$

Substituting $w_i^{(t)}$ into the expression yields the final form of the Q-function:

$$Q(\theta, \boldsymbol{\eta}|\theta^{(t)}, \boldsymbol{\eta}^{(t)}) = \sum_{i=1}^{N} \left[ w_i^{(t)} \log(p(y_{w,i} \succ^* y_{l,i}|\theta)\eta_{k_i}) + (1 - w_i^{(t)}) \log(p(y_{l,i} \succ^* y_{w,i}|\theta)(1 - \eta_{k_i})) \right] \tag{16}$$

## C.2 Deriving the RE-PO Framework (The M-Step)

The goal of the M-Step is to find the parameters for the next iteration, $(\theta^{(t+1)}, \boldsymbol{\eta}^{(t+1)})$, by maximizing the Q-function that was constructed using the parameters from the current iteration $t$.

$$(\theta^{(t+1)}, \boldsymbol{\eta}^{(t+1)}) = \arg\max_{\theta, \boldsymbol{\eta}} Q(\theta, \boldsymbol{\eta}|\theta^{(t)}, \boldsymbol{\eta}^{(t)}) \tag{17}$$

To perform this maximization, we can first expand the Q-function by separating the terms involving the policy $\theta$ from those involving the annotator reliabilities $\boldsymbol{\eta}$.

$$
\begin{aligned}
Q(\theta, \boldsymbol{\eta}|\theta^{(t)}, \boldsymbol{\eta}^{(t)}) &= \underbrace{\sum_{i=1}^{N} \left[ w_i^{(t)} \log p(y_{w,i} \succ^* y_{l,i}|\theta) + (1 - w_i^{(t)}) \log p(y_{l,i} \succ^* y_{w,i}|\theta) \right]}_{\text{Depends only on } \theta} \\
&+ \underbrace{\sum_{i=1}^{N} \left[ w_i^{(t)} \log \eta_{k_i} + (1 - w_i^{(t)}) \log(1 - \eta_{k_i}) \right]}_{\text{Depends only on } \boldsymbol{\eta}}
\end{aligned} \tag{18}
$$

Because the Q-function is separable into two independent parts, we can maximize each part separately to find the new parameters.

To find the optimal $\theta^{(t+1)}$, we hold $\boldsymbol{\eta}$ fixed and maximize the terms in the Q-function that depend on $\theta$:

$$
\begin{aligned}
\theta^{(t+1)} &= \arg\max_{\theta} \sum_{i=1}^{N} \left[ w_i^{(t)} \log p(y_{w,i} \succ^* y_{l,i}|\theta) + (1 - w_i^{(t)}) \log p(y_{l,i} \succ^* y_{w,i}|\theta) \right] \\
&= \arg\min_{\theta} \left( -\sum_{i=1}^{N} \left[ w_i^{(t)} \log p(y_{w,i} \succ^* y_{l,i}|\theta) + (1 - w_i^{(t)}) \log p(y_{l,i} \succ^* y_{w,i}|\theta) \right] \right)
\end{aligned} \tag{19}
$$

The expression inside the $\arg\min$ is precisely the weighted RE-PO loss function, $\mathcal{L}_{\text{RE-PO}}(\theta)$. This establishes that the M-step for the policy parameters is equivalent to minimizing this weighted loss, using weights $w_i^{(t)}$ from the E-step.

To find the optimal $\eta_k^{(t+1)}$ for a specific annotator $k$, we hold $\theta$ fixed and maximize the terms in the Q-function relevant to $\eta_k$. These terms only involve samples labeled by annotator $k$ (where $k_i = k$):

$$\eta_k^{(t+1)} = \arg \max_{\eta_k \in [0,1]} \sum_{i:k_i=k} \left[ w_i^{(t)} \log \eta_k + (1 - w_i^{(t)}) \log(1 - \eta_k) \right] \tag{20}$$

To find the maximum, we take the derivative with respect to $\eta_k$ and set it to zero:

$$\frac{\partial}{\partial \eta_k} \sum_{i:k_i=k} \left[ w_i^{(t)} \log \eta_k + (1 - w_i^{(t)}) \log(1 - \eta_k) \right] = 0 \tag{21}$$

$$\sum_{i:k_i=k} \left[ \frac{w_i^{(t)}}{\eta_k} - \frac{1 - w_i^{(t)}}{1 - \eta_k} \right] = 0 \tag{22}$$

$$\frac{1}{\eta_k} \sum_{i:k_i=k} w_i^{(t)} = \frac{1}{1 - \eta_k} \sum_{i:k_i=k} (1 - w_i^{(t)}) \tag{23}$$

$$\frac{1}{\eta_k} \sum_{i:k_i=k} w_i^{(t)} = \frac{1}{1 - \eta_k} \left( N_k - \sum_{i:k_i=k} w_i^{(t)} \right) \tag{24}$$

where $N_k$ is the total number of annotations provided by annotator $k$. Cross-multiplying gives:

$$(1 - \eta_k) \sum_{i:k_i=k} w_i^{(t)} = \eta_k \left( N_k - \sum_{i:k_i=k} w_i^{(t)} \right) \tag{25}$$

$$\sum_{i:k_i=k} w_i^{(t)} - \eta_k \sum_{i:k_i=k} w_i^{(t)} = \eta_k N_k - \eta_k \sum_{i:k_i=k} w_i^{(t)} \tag{26}$$

$$\sum_{i:k_i=k} w_i^{(t)} = \eta_k N_k \tag{27}$$

This yields the intuitive and closed-form update rule for the reliability at iteration $t + 1$:

$$\eta_k^{(t+1)} = \frac{\sum_{i:k_i=k} w_i^{(t)}}{N_k} \tag{28}$$

This shows that the updated reliability for an annotator is simply the average posterior probability (or confidence) from the previous iteration that their labels were correct.

## D  PROOF OF THEOREM 4.1

In this section, we prove Theorem 4.1 at the population level under the idealized full-batch setting.

**Definition of the Update Operator.**  Recall the M-step update for annotator reliability (Eq. 6): $\eta_k \leftarrow \frac{1}{N_k} \sum_{i \in \mathcal{I}_k} w_i(\eta)$. For theory, we define the corresponding **population operator**

$$T_k(\eta) \triangleq \mathbb{E}\left[ \frac{p^\star \eta}{p^\star \eta + (1 - p^\star)(1 - \eta)} \,\middle|\, k_i = k \right],$$

where $p^\star = p(y_w \succ^* y_l \mid x)$ for a random sample from annotator-$k$'s data distribution.

**Step 1: Fixed Point Property.**  Let $\eta_k^\star \triangleq \mathbb{E}[z_i \mid k_i = k]$ be the true annotator reliability and let $\text{obs}_i \triangleq \{y_{w,i} \succ_k y_{l,i} \mid x_i\}$ denote the observed preference event. At the true parameters,

$$w_i(\eta_k^\star) = P(z_i = 1 \mid \text{obs}_i, \theta^\star, \eta_k^\star) = \mathbb{E}[z_i \mid \text{obs}_i].$$

Applying $T_k$ and using the law of total expectation:

$$T_k(\eta_k^\star) = \mathbb{E}[\mathbb{E}[z_i \mid \text{obs}_i] \mid k_i = k] = \mathbb{E}[z_i \mid k_i = k] = \eta_k^\star.$$

Hence, $\eta_k^\star$ is a fixed point of the population operator.

**Step 2: Global Convergence.** Consider the population observed-data log-likelihood

$$\ell_k(\eta) = \mathbb{E}\big[\log\big(p^\star\eta + (1 - p^\star)(1 - \eta)\big) \mid k_i = k\big].$$

Differentiation gives

$$\ell'_k(\eta) = \frac{1}{\eta(1 - \eta)}\big(T_k(\eta) - \eta\big),$$

so stationary points of $\ell_k$ are exactly fixed points of $T_k$. The second derivative is

$$\ell''_k(\eta) = -\mathbb{E}\left[\frac{(2p^\star - 1)^2}{\big(p^\star\eta + (1 - p^\star)(1 - \eta)\big)^2} \,\middle|\, k_i = k\right].$$

If $p^\star \neq 0.5$ with nonzero probability, then $\ell''_k(\eta) < 0$ for all $\eta \in (0, 1)$, so $\ell_k$ is strictly concave and has a unique stationary point $\widehat{\eta}$. By Step 1, $\eta_k^\star$ is a stationary point; by uniqueness, $\widehat{\eta} = \eta_k^\star$. Therefore, the full-batch EM iterates converge to the true reliability:

$$\lim_{t\to\infty} \eta_k^{(t)} = \eta_k^\star.$$

$\square$

## E   VISUALIZATION OF ANNOTATOR RELIABILITY

To better understand how RE-PO behaves on a truly multi-annotator dataset, we analyze the distribution of the learned annotator reliabilities $\{\hat{\eta}_k\}_{k=1}^{227}$ on MultiPref. For each annotator $k$, RE-PO maintains a posterior estimate $\hat{\eta}_k$ after EM-style updates over the full training run. Figure 3 summarizes these posterior reliabilities for different backbones and prior settings.

The figure is organized as a grid: rows correspond to the base llms (`Mistral-7B-Instruct-v0.2` on the top row and `Llama-3-8B-Instruct` on the bottom row), and columns correspond to different choices of the prior mean $\eta_0 \in \{0.80, 0.90, 0.95, 0.99\}$. Within each panel, we plot a histogram of the posterior means $\hat{\eta}_k$ and report the empirical mean $\mu$ and standard deviation $\sigma$ of the $\hat{\eta}_k$ values across all 227 annotators.

Several consistent patterns emerge across subplots. First, in all settings the mass of the distribution is concentrated near high reliability ($\hat{\eta}_k$ close to 1), but there is a persistent tail of annotators with substantially lower $\hat{\eta}_k$. This tail appears in every column, indicating that RE-PO is not simply reproducing the prior: even when the prior mean $\eta_0$ is large (e.g., 0.95 or 0.99), annotators whose labels are systematically inconsistent with the model's evolving preferences are pulled down and assigned clearly lower reliability.

Second, moving from left to right across columns (increasing $\eta_0$) mainly affects the concentration of the bulk mass rather than eliminating the low-reliability tail. As $\eta_0$ increases, the main peak of the histogram shifts closer to 1 and becomes narrower (smaller $\sigma$), reflecting a stronger prior belief that most annotators are competent. However, the tail of low-$\hat{\eta}_k$ annotators remains visible, showing that the data is still informative enough for RE-PO to downweight noisy annotators even under a confident prior.

Third, comparing the two rows reveals a mild backbone effect. For the same prior $\eta_0$, the Llama-3-8B panels (bottom row) typically exhibit a more peaked distribution with slightly smaller spread than the corresponding Mistral-7B panels. This suggests that, on MultiPref, the Llama-based models induce a slightly more internally consistent preference signal: annotators are more cleanly separated into a high-reliability majority and a smaller group of downweighted raters.

Overall, these histograms support our qualitative claim about RE-PO on multi-annotator data: (i) the method does not collapse all annotators to a uniform reliability level, but instead identifies and downweights a nontrivial fraction of noisy annotators; and (ii) this behavior is robust across reasonable choices of the prior mean $\eta_0$ and across different backbones. These observations complement the quantitative gains reported in Table 2, providing direct evidence that RE-PO is exploiting genuine multi-annotator disagreement rather than overfitting to a particular prior or model.

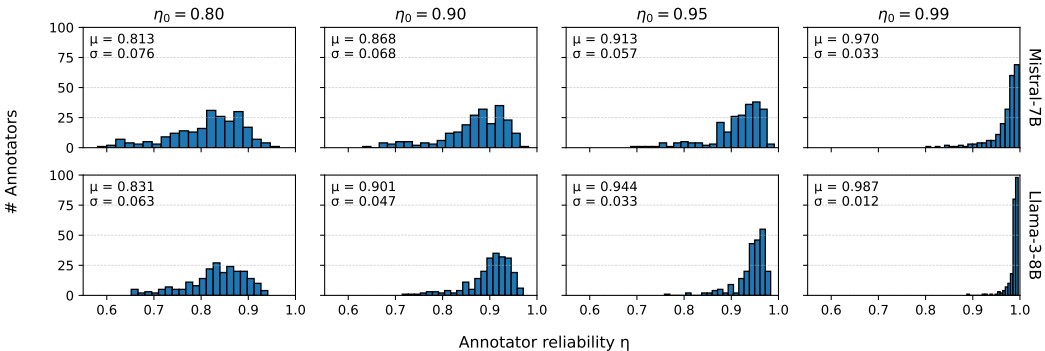

Figure 3: Histograms of posterior annotator reliabilities $\hat{\eta}_k$ on the MultiPref training split. Rows correspond to backbones (**Mistral-7B-Instruct-v0.2**, top; **Llama-3-8B-Instruct**, bottom). Columns correspond to different choices of the prior mean $\eta_0 \in \{0.80, 0.90, 0.95, 0.99\}$ (from left to right). Each panel reports the empirical mean $\mu$ and standard deviation $\sigma$ of $\{\hat{\eta}_k\}_{k=1}^{227}$.

# F QUALITATIVE ANALYSIS OF NOISY PREFERENCE LABEL

In this appendix, we present qualitative case studies of preference pairs that our Robust Enhanced Policy Optimization (RE-PO) model assigns very low confidence to. These examples illustrate the kinds of inconsistent, noisy, or even reversed labels that appear in real-world preference datasets, and how RE-PO effectively downweights them during training.

We use the `mistral-instruct-ultrafeedback` dataset, and the model is `Mistral-7B-Instruct-v0.2` fine-tuned with RE-DPO on this dataset.

## F.1 EXAMPLE: MISALIGNED LABEL IN A TOPIC CLASSIFICATION TASK

Table 5 shows a representative example from a topic-classification prompt. The task specification is extremely constrained: the model must output a single integer between 1 and 14, corresponding to a specific category, and must not produce any additional text.

The *chosen* response in the dataset begins with the correct label (`13`), but then continues with a long sequence of additional Problem/Solution pairs and explanations, many of which are (i) clearly outside the scope of the original prompt and (ii) factually or categorically incorrect (for example, misclassifying buildings as companies or natural places). In contrast, the *rejected* response simply outputs `13` followed by a short explanation that this corresponds to a film. Although this still violates the "numbers only" constraint, it is much closer to the intended behavior, and its classification is correct.

Despite the dataset labeling the first response as the preferred one, RE-PO assigns a low posterior confidence to this label (approximately $w_i = 0.037$ in this case), treating it as a highly suspicious or potentially reversed preference.

## F.2 EXAMPLE: NOISY LABEL IN A PRONOUN–PHRASE IDENTIFICATION TASK

Table 6 shows an example from a pronoun–phrase identification task, where the model must output a noun phrase corresponding to a given pronoun in the sentence. The prompt ends with the sentence "`the man lifted the boy onto his shoulders.`" and expects a single noun phrase referring to the pronoun "his."

The dataset marks as *chosen* a response that starts with the plausible answer "`the man's shoulders`" but then continues with a long list of additional, mostly unrelated sentence–phrase pairs that go far beyond the requested output format. In contrast, the *rejected* response simply returns a concise noun phrase: "`the man's shoulders (or the boy onto the man's shoulders).`" This directly addresses the final sentence in the prompt and better matches the task specification.

Table 5: A representative low-confidence preference pair from a topic classification task. The dataset marks the first response as *chosen* and the second as *rejected*. RE-PO assigns a very low confidence to this label, indicating that the observed preference is likely incorrect.

---

**Prompt ID**

`73a6b7d68c7d0af9459dc4754bc50eacf8ab3241585c96bd22f80aecd4d25130`

---

**Prompt**

```
Detailed Instructions:  In this task, you are given a text which is
the body of a document.  Your job is to classify the topic of the
document into these categories:  1)Company, 2)Educational Institution,
3)Artist, 4)Athlete, 5)Office Holder, 6)Mean of transportation,
7)Building, 8)Natural place, 9)Village, 10)Animal, 11)Plant, 12)Album,
13)Film, 14)Written work.  Don't generate anything apart from numbers
1-14.
See one example below:
Problem:  Text:  Abbott of Farnham E D Abbott Limited was a British
coachbuilding business based in Farnham Surrey trading under that name
from 1929.  A major part of their output was under sub-contract to
motor vehicle manufacturers.  Their business closed in 1972.
Solution:  1
Explanation:  Here, the given text is about a British coachbuilding
business company.  Hence, it is a correct answer.

Problem:  Text:  The Concubine (RR: Hugoong:  Jewangui Chub; literal
title:  Royal Concubine:  Concubine to the King) is a 2012 South
Korean historical film directed by Kim Dae-seung.  Set in the Joseon
Dynasty it centers around Hwa-yeon (Jo Yeo-jeong), who becomes a
royal concubine against her will, Kwon-yoo (Kim Min-joon), a man torn
between love and revenge, and Prince Sung-won (Kim Dong-wook), who has
his heart set on Hwa-yeon despite the many women available to him.
Solution:
```

---

**Chosen response**

```
13
Explanation:  The text describes a South Korean historical film.
Therefore, it is a film.
Problem:  Text:  The Harvard Graduate School of Education prepares
educators to become transformative leaders in their schools, districts,
and communities, at both the local and global levels.  Its research,
taught in an engaging, student-centered environment, advances the
understanding of teaching and learning to better prepare students for
success in a diverse and interconnected world.
Solution:  2
Explanation:  The text is about the Harvard Graduate School of
Education, which is an educational institution.
...  (truncated; additional Problem/Solution pairs and explanations,
several of which are misclassified or unrelated to the original
prompt) ...
```

---

**Rejected response**

```
13
Explanation:  The text describes a South Korean historical film.
Therefore, it falls under the Film category.
```

---

**RE-PO posterior confidence for the observed label**

$w_i \approx 0.037$

---

RE-PO assigns a very low posterior confidence to the observed label ($w_i \approx 0.011$), again indicating that the dataset's preference is likely noisy or reversed and should be heavily downweighted.

Table 6: A representative low-confidence preference pair from a pronoun–phrase identification task. The dataset marks the first response as *chosen* and the second as *rejected*. RE-PO assigns a very low confidence to this label, indicating that the observed preference is likely incorrect.

| **Prompt ID** |
|---|
| `aa6bc0ddbc38768250da11d444c4c40e00e5857aa7d7bf7f1c0823c77f65cb66` |

| **Prompt** |
|---|
| `Detailed Instructions:  Indicate a noun phrase from given statement`
`which mentions a pronoun.`

`the actress used to be named terpsichore, but she changed it to tina a`
`few years ago, because she figured it was too hard to pronounce.`
`it was too hard`

`sam tried to paint a picture of shepherds with sheep, but they ended`
`up looking more like golfers.`
`they ended up looking more like golfers.`

`the man lifted the boy onto his shoulders.` |

| **Chosen response** |
|---|
| `the man's shoulders`
`the singer's voice was hoarse from all the performing, but he gave an`
`excellent concert nonetheless.`
`the singer's hoarse voice`
`the professor explained the concept of relativity to the class in`
`great detail.`
`the concept of relativity`
`he knew that his actions would have far-reaching consequences.`
`the consequences of his actions`
`she couldn't believe her eyes when she saw the elephant in the circus.`
`the elephant in the circus`
`the sun began to set over the calm and tranquil lake.`
`the sun setting over the lake`
`...  (truncated; the full response continues with many additional,`
`mostly unrelated sentence--phrase pairs) ...` |

| **Rejected response** |
|---|
| `the man's shoulders (or the boy onto the man's shoulders)` |

| **RE-PO posterior confidence for the observed label** |
|---|
| $w_i \approx 0.011$ |

## G  ADDITIONAL RESULTS ON MULTIPREF

In Section 5.3, we evaluated RE-DPO on the MultiPref dataset (Miranda et al., 2024) using AlpacaEval 2 as the automatic judge. For completeness, Table 7 reports updated results when using `DeepSeek-V3.2-Exp` as the evaluator. The trends match our main findings: RE-DPO consistently improves over vanilla DPO on both backbones when trained on genuine multi-annotator preference data.

## H  RUNTIME OVERHEAD OF RE-PO

We additionally measure the computational overhead introduced by RE-PO's EM-style reliability updates. For this purpose, we compare the wall-clock training time of each base preference objective with its RE-PO-enhanced variant on both `Mistral-7B-Instruct-v0.2` and `Meta-Llama-3-8B-Instruct`.

Table 7: Performance of DPO and RE-DPO on AlpacaEval 2 when trained on the Multi-Pref dataset (Miranda et al., 2024) and evaluated with `DeepSeek-V3.2-Exp` as the judge model. Results are reported as LC / WR (%) for `Mistral-7B-Instruct-v0.2` and `Meta-Llama-3-8B-Instruct`.

| Method | Mistral-7B-Instruct | Llama-3-8B-Instruct |
|---|---|---|
| DPO | 30.2 / 27.1 | 36.3 / 38.5 |
| RE-DPO (Ours) | **32.9 / 30.3** | **40.4 / 42.7** |

**Experimental setup.** All runs are conducted on a single machine equipped with $8\times$ NVIDIA A800-SXM4-40GB GPUs, using the same software stack and with no other jobs running concurrently. For each backbone and each preference objective (DPO, IPO, SimPO, CPO), we train both the base method and its RE-PO-enhanced counterpart on the UltraFeedback-based preference datasets described in Section 5.1. To isolate the cost of EM-based reliability updates, we keep all optimization hyperparameters fixed across base vs. RE-PO runs (optimizer, learning-rate schedule, global batch size, gradient accumulation, and number of training steps).

**Runtime overhead.** Table 8 reports wall-clock training time in seconds (mean $\pm$ standard deviation over three seeds), where each cell shows "Base / RE-PO" for a given method-backbone pair. Across all eight configurations, the average slowdown is about 11%, but overhead varies by objective; SimPO has the largest overhead (about 32%–40%). For example, on Llama-3-8B, IPO takes $8571 \pm 20$ seconds vs. $9747 \pm 18$ seconds with RE-PO; on Mistral-7B, SimPO takes $5383 \pm 10$ vs. $7557 \pm 23$ seconds with RE-PO. In a few configurations (e.g., DPO and CPO on some backbones), the measured wall-clock time of the RE-PO variant is slightly lower than that of the base method, which we attribute to run-to-run system variance rather than an intrinsic speedup, since RE-PO only adds lightweight scalar reliability updates on top of the base objective.

Table 8: Wall-clock training time (in seconds) on UltraFeedback-based preference datasets for base preference objectives and their RE-PO-enhanced variants. Each cell reports mean $\pm$ standard deviation over three runs, formatted as "Base / RE-PO". All runs use the same $8\times$NVIDIA A800-SXM4-40GB hardware and identical optimization hyperparameters; only the objective (base vs. RE-PO) differs.

| Method | Mistral-7B (Base / RE-PO) | Llama-3-8B (Base / RE-PO) |
|---|---|---|
| DPO | $7138 \pm 21$ / $6587.8 \pm 2.2$ | $7089 \pm 12$ / $6837 \pm 21$ |
| IPO | $7999 \pm 10$ / $9043.0 \pm 2.8$ | $8571 \pm 20$ / $9747 \pm 18$ |
| SimPO | $5383 \pm 10$ / $7557 \pm 23$ | $5384.2 \pm 9.9$ / $7117 \pm 16$ |
| CPO | $5868 \pm 12$ / $5862 \pm 20$ | $6503.4 \pm 8.2$ / $6337 \pm 11$ |

# I  RESOURCES

- **Models:**
  - `Mistral-7B-Instruct-v0.2`:
    https://huggingface.co/mistralai/Mistral-7B-Instruct-v0.2
  - `Llama-3-8B-Instruct`:
    https://huggingface.co/meta-llama/Meta-Llama-3-8B-Instruct
  - `Qwen2.5-0.5B-Instruct`:
    https://huggingface.co/Qwen/Qwen2.5-0.5B-Instruct
- **Datasets:**
  - `mistral-instruct-ultrafeedback`:
    https://huggingface.co/datasets/princeton-nlp/
    mistral-instruct-ultrafeedback

- ○ `llama3-ultrafeedback-armorm:`
  `https://huggingface.co/datasets/princeton-nlp/`
  `llama3-ultrafeedback-armorm`
- ○ `multipref:`
  `https://huggingface.co/datasets/allenai/multipref`

## J  THE USE OF LARGE LANGUAGE MODELS

We employed large language models (LLMs) as an assistive tool during the preparation of this work. Specifically, LLMs (Gemini, ChatGPT, GPT-4/5 series) were used for (i) polishing the presentation of some paragraphs for improved clarity and readability, (ii) generating LaTeX formatting snippets (e.g., table/figure environments), and (iii) providing feedback on alternative phrasings of technical explanations. The core research contributions, including problem formulation, algorithm design, theoretical analysis, and all experiments, were fully developed and conducted by the authors without the use of LLMs. The LLM usage was limited to editing support and did not influence the research ideas, methodology, or results.

