# OpenReview forum: "RE-PO: Robust Enhanced Policy Optimization as a General Framework for LLM Alignment"
_ICLR.cc/2026/Conference — ICLR 2026 Poster_

### Official Review · Reviewer_tvbG · 2025-10-28

**Soundness:** 3
**Presentation:** 3
**Contribution:** 3
**Rating:** 6
**Confidence:** 4

**Summary:**

This paper introduces Robust Preference Optimization (RPO), a novel method designed to address the challenge of label noise in Large Language Model (LLM) alignment. The authors correctly identify that existing alignment methods like DPO, IPO, and others operate under the strong and often flawed assumption that all human preference labels are correct. RPO reframes the problem by treating the true preference as a latent variable and employs an Expectation-Maximization (EM) framework to jointly infer the correctness of labels (E-step) and update the model policy along with annotator reliability parameters (M-step). A key strength is its design as a meta-framework, making it compatible with various existing preference loss functions. Comprehensive experiments on models like Mistral-7B and Llama-3-8B across benchmarks like AlpacaEval 2 and ArenaHard show consistent and significant performance gains when RPO is applied to several base algorithms (DPO, IPO, SimPO, CPO). The paper also provides theoretical guarantees, proving that under idealized conditions, RPO can recover the true reliability of annotators.

**Strengths:**

1. The core idea of applying an EM-based, soft-labeling approach to LLM preference alignment is highly innovative. Moving from hard, assumed-correct labels to a probabilistic model that explicitly accounts for noise is a significant conceptual and practical advancement.

2. The formulation of RPO as a wrapper that can enhance a wide range of existing alignment methods (demonstrated with DPO, IPO, SimPO, CPO) is a major strength. This greatly increases the potential impact and applicability of the work.

3. The paper is not merely empirical; it provides a solid theoretical foundation. The derivation of the EM algorithm for this context and, crucially, the proof of convergence to the true annotator reliability (Theorem 4.2) lend significant credibility to the method.

4. The experimental section is thorough, validating the method's effectiveness across two different base models, two challenging benchmarks, and four distinct alignment algorithms. The consistent improvements are compelling evidence for RPO's utility. The authors thoughtfully address the computational challenges of a full-batch EM algorithm by proposing a practical mini-batch training scheme with an Exponential Moving Average (EMA) for updating annotator reliabilities, making the method feasible for large-scale training.

**Weaknesses:**

1.  A notable weakness is the experimental setup where all preference data is modeled as coming from a single, virtual annotator (K=1). While the justification in a footnote is reasonable, it undermines the validation of one of RPO's core capabilities: distinguishing between multiple annotators of varying reliability. The empirical results primarily demonstrate noise robustness in general, but not necessarily the multi-annotator aspect.

2. The ablation study reveals that performance is sensitive to the initial annotator reliability (η₀) and the EMA momentum (α). This sensitivity could pose a non-trivial tuning burden in practice, especially in real-world scenarios where the optimal values are not known a priori.

3. The experiments rely on existing datasets and synthetic noise injection. While valuable, this does not fully capture the complexity and structure of noise found in real-world, multi-annotator preference data. Validation on a dataset with genuine, traceable multi-annotator labels would be stronger.

4. The EM procedure, with its additional forward passes to compute confidences (w_i) and updates for η, inevitably introduces computational overhead compared to the base algorithms. While likely acceptable given the performance gains, a brief discussion of the training time/ cost increase would be helpful for practitioners.

**Questions:**

1. How would RPO perform in a setting with a dataset containing multiple annotators with widely different and unknown reliabilities? Are there plans to validate the method on a dataset with genuine, traceable multi-annotator labels (e.g., from a crowdsourcing platform)?

2. Theorem 4.2 assumes full-batch updates and a perfectly calibrated model. In the practical mini-batch setting with stochastic gradient descent, have you observed any issues with the convergence or stability of the EM procedure, particularly for the annotator reliability estimates?

3. The experiments are conducted on 7B and 8B parameter models. Do you anticipate any challenges (e.g., memory, convergence) in scaling RPO to much larger models (e.g., 70B+ parameters), and have you conducted any preliminary experiments in this direction?

---

> ### Author Response · Authors · 2025-11-22
> **Response to Reviewer tvbG (1/5)**
>
> Thank you for your constructive comments and feedback! We hope the following response can address your concerns.
>
> ---
>
> > Weakness 1: A notable weakness is the experimental setup where all preference data is modeled as coming from a single, virtual annotator (K=1). While the justification in a footnote is reasonable, it undermines the validation of one of RPO's core capabilities: distinguishing between multiple annotators of varying reliability ...
> >
> > Weakness 3: The experiments rely on existing datasets and synthetic noise injection. While valuable, this does not fully capture the complexity and structure of noise found in real-world, multi-annotator preference data. Validation on a dataset with genuine, traceable multi-annotator labels would be stronger.
> >
> > Question 1: How would RPO perform in a setting with a dataset containing multiple annotators with widely different and unknown reliabilities? Are there plans to validate the method on a dataset with genuine, traceable multi-annotator labels (e.g., from a crowdsourcing platform)?
>
> We agree that validating RPO on datasets with genuine multi-annotator disagreement is important. To validate RPO under realistic multi-annotator dataset, we perform experiments using Multipref dataset [1]. We compare DPO and R-DPO (Ours) on Mistral-7B-Instruct-v0.2 and Llama-3-8B-Instruct. Results in Rebuttal Table 1 show that RPO performs better on multi-annotator dataset. These new results are included in sec 5.3 in the revised manuscript.
>
> - **Dataset description.** Each preference pair in Multipref is annotated multiple times by distinct human annotators, and the dataset covers a broad range of open-ended prompts and response types. In our experiments we use the official training split containing **227 unique human annotators**.
>
> - **Experimental setup and results.** We train vanilla DPO and our R-DPO on MultiPref for both Mistral-7B and Llama-3-8B, and then evaluate the resulting models on AlpacaEval 2. As shown in Rebuttal Table 1, R-DPO consistently outperforms vanilla DPO with multi-annotator dataset: for Llama-3-8B, AlpacaEval LC improves from ** 36.7** to **41.1** and WR from **39.3** to **44.4**; for Mistral-7B, LC improves from **28.8** to **31.8** and WR from **26.4** to **28.8**.
>
> _**Rebuttal Table 1**. The performance of different models on AlpacaEval 2._
>
> | Model                | LC (%) | WR (%) |
> | :------------------- | ------ | ------ |
> | Mistral-DPO          | 28.8   | 26.4   |
> | Mistral-R-DPO (Ours) | **31.8**   | **28.8**   |
> | Llama-DPO            | 36.7   | 39.3   |
> | Llama-R-DPO (Ours)   | **41.1**   | **44.4**   |
>
> [1] Miranda, Lester James Validad, et al. "Hybrid preferences: Learning to route instances for human vs. AI feedback." ACL 2025.

---

> ### Author Response · Authors · 2025-11-22
> **Response to Reviewer tvbG (2/5)**
>
> > Weakness 2: The ablation study reveals that performance is sensitive to the initial annotator reliability (η₀) and the EMA momentum (α). This sensitivity could pose a non-trivial tuning burden in practice, especially in real-world scenarios where the optimal values are not known a priori.
>
> We agree that $\eta_0$ and $\alpha$ are important hyperparameters. However, our ablations suggest that (i) both are reasonably robust within a broad *reasonable* range, (ii) $\alpha$ can be fixed once and reused across datasets and base algorithms, and (iii) even in suboptimal settings, RPO typically still outperforms the non-RPO baselines. We will add practical default choices and ranges to reduce tuning burden in real-world use.
>
> - **Robustness within a reasonable range.** One of the goal of our ablation in Sec. 5.4 was to stress-test $\eta_0$ $\alpha$, including quite extreme values. In these extremes, the EM updates can indeed become less stable and performance drops, which we now state more clearly. However, within a broad and practically reasonable range, the curves are relatively flat: performance varies only slightly and remains **consistently above** the corresponding base methods without RPO.
>
> - **$\alpha$ is easy to fix and does not depend strongly on the dataset.** In all our experiments, we found that a single EMA momentum value $\alpha=0.1$ works well across different datasets (UltraFeedback, MultiPref), base objectives (DPO, IPO, SimPO, CPO), and backbones (Mistral-7B, Llama-3-8B). For a standard EMA range (e.g., $\alpha=0.1\sim 0.01$), we observed stable behavior across the board. In the revision, we will explicitly recommend a default α and range, so practitioners can simply reuse these values without extensive tuning.
>
> - $\eta_0$ is interpretable and robust in practice. $\eta_0$ controls how strongly we trust annotators *a priori*. For moderate $\eta_0$ values around our default choice, RPO’s performance is quite stable and, again, remains better than the base objectives. Because $\eta_0$ has a clear interpretation (“how reliable is an average annotator before seeing data?”), we can provide simple rules-of-thumb (e.g., starting from a neutral prior and adjusting slightly if one expects more or less noise).

---

> > ### Author Response · Authors · 2025-11-22
> > **Response to Reviewer tvbG (3/5)**
> >
> > > Weakness 4: The EM procedure, with its additional forward passes to compute confidences and updates for η, inevitably introduces computational overhead compared to the base algorithms. While likely acceptable given the performance gains, a brief discussion of the training time/ cost increase would be helpful for practitioners.
> >
> > Thanks for the suggestion! We agree that it is important to quantify the runtime impact of our EM-style reliability updates. To address this concern, we ran additional controlled experiments comparing the wall-clock training time of each base preference objective with its RPO-enhanced variant on both Mistral-7B and Llama-3-8B. These results are included in Appendix H in the revised manuscript.
> >
> > - **Experimental setup and hardware.** All experiments were conducted on a single machine equipped with 8× NVIDIA A800-SXM4-40GB GPUs (40 GB per GPU), using the same software stack and with no other jobs running concurrently. For each backbone (Mistral-7B-Instruct-v0.2 and Llama-3-8B-Instruct) and each preference objective (DPO, IPO, SimPO, CPO), we train both the base method and its RPO-enhanced counterpart on the UltraFeedback dataset.
> >
> > - **Controlled variables.** To isolate the cost of EM-based reliability updates, we keep all other settings fixed across base vs. RPO runs (e.g., the optimizer (AdamW), global batch size and gradient accumulation, etc.). The only difference is whether RPO is applied.
> >
> > - **Runtime overhead.** Rebuttal Table 2 reports wall-clock training time in seconds (mean ± standard deviation over 3 seeds), where each cell shows “Base / RPO” for a given method–backbone pair. Across all 8 configurations, RPO stays within roughly **20%** of the corresponding base method, with an average slowdown of ≈11%. For example, on Llama-3-8B, IPO takes 8571 ± 20 seconds vs. 9747 ± 18 seconds with RPO. Given RPO's improvement in performance on benchmarks, this range of additional computational time is acceptable.
> >
> > Note that in a few configurations (e.g., DPO and CPO on some backbones), the measured wall-clock time of the RPO variant is slightly better than that of the base method. We **do not** interpret this as an inherent speedup from RPO. Since RPO only adds very lightweight scalar reliability updates on top of the base objective, these seed- and padding-induced fluctuations can easily outweigh the marginal extra computation, occasionally making RPO appear marginally faster.
> >
> > - **Training stability.** Across all experiments reported in the paper, we did not observe any instability specific to RPO: the training loss curves and held-out metrics evolve smoothly and converge in a similar number of steps as their corresponding base methods. Together with the $\eta_0 / \alpha$ ablation in Sec. 5.3, this suggests that introducing EM-style reliability updates does not make optimization more brittlein practice.
> >
> > _**Rebuttal Table 2**. Training time in seconds on UltraFeedback (mean ± std over 3 runs). Each cell reports the training time for the base preference objective and its RPO-enhanced variant as “Base / RPO”.  All runs use the same hardware (8× NVIDIA A800-SXM4-40GB GPUs) and identical optimization hyperparameters; only the objective (base vs. RPO) differs._
> >
> > | Method | Mistral-7B (Base / RPO)   |  Llama-3-8B (Base / RPO)  |
> > | :----: | ------------------------- | ------------------------- |
> > |  DPO   | $7138\pm21$ / $6588\pm2$  | $7089\pm12$ / $6837\pm21$ |
> > |  IPO   | $7999\pm10$ / $9043\pm3$  | $8571\pm20$ / $9747\pm18$ |
> > | SimPO  | $5383\pm10$ / $7557\pm23$ | $5384\pm10$ / $7117\pm16$ |
> > |  CPO   | $5868\pm12$ / $5862\pm20$ | $6503\pm8$  / $6337\pm11$ |

---

> > > ### Author Response · Authors · 2025-11-22
> > > **Response to Reviewer tvbG (4/5)**
> > >
> > > > Question 2: Theorem 4.2 assumes full-batch updates and a perfectly calibrated model. In the practical mini-batch setting with stochastic gradient descent, have you observed any issues with the convergence or stability of the EM procedure, particularly for the annotator reliability estimates?
> > >
> > > Thanks for the meaningful advice! Theorem 4.2 analyzes an idealized setting (full-batch, perfectly calibrated model). In practice we use mini-batch SGD with EMA-style updates and well-aligned base models, and we **do not** observe convergence or stability issues for the EM procedure or the reliability estimates under propoer hyperparameter settings. We also discuss this problem in Sec. 4 in the revised manuscript.
> > >
> > > - **What Theorem 4.2 actually covers (and what it doesn’t).** Theorem 4.2 is an idealized result. Its role is to show that, under these conditions, RPO will not distort a well-calibrated model and will recover the true annotator reliabilities. It is **not** meant as a full convergence proof for practical mini-batch training. In the revision, we will make this scope explicit.
> > >
> > > - **How we approximate these assumptions in practice.**
> > >    - **Full-batch vs. mini-batch.** Instead of a true full-batch E-step, we use an exponential moving average (EMA) over mini-batches to update the reliability parameters. We also train with a reasonably large global batch size (64 in our main experiments), which we found to be sufficient for stable behavior. Intuitively, EMA over many mini-batches plays a similar role to a full-batch update averaged over the dataset.
> > >    - **Perfect calibration vs. realistic base models.** We start from standard instruction-tuned base models (Mistral-7B-Instruct-v0.2 and Llama-3-8B-Instruct), , which are widely used as base models for preference optimization and alignment in recent work and libraries [2, 3, 4]. These models are already reasonably aligned with human preferences. They are not perfectly calibrated, but in practice they provide informative preference probabilities from the very beginning of training.
> > >
> > > - **Empirical evidence on convergence and stability.** In all our experiments, when hyperparameters are in the reasonable ranges we report in Sec. 5.3, we **do not** observe instability:
> > >    - The annotator reliabilities evolve smoothly and then stabilize, rather than bouncing wildly between batches.
> > >    - The training loss curves for RPO and the base methods are smooth and decrease in a similar way.
> > >    - The AlpacaEval 2 scores improve steadily rather than oscillating.
> > >    This suggests that, under standard mini-batch SGD with EMA, the EM procedure behaves well in practice.
> > >
> > > - **Limitations and future work.** We agree that a full theoretical treatment of the stochastic mini-batch EM setting (without assuming perfect calibration) would be valuable. We will add this as a natural future direction in the conclusion section.
> > >
> > > [2] Meng, Y., Xia, M., and Chen, D. (2024). SimPO: Simple Preference Optimization with a Reference-Free Reward. arXiv:2405.14734.
> > >
> > > [3] Wu, Y., Sun, Z., Yuan, H., Ji, K., Yang, Y., and Gu, Q. (2024). Self-Play Preference Optimization for Language Model Alignment. arXiv:2405.00675.
> > >
> > > [4] Just, H. A., Jin, M., Sahu, A. K., Phan, H., and Jia, R. (2024). Data-Centric Human Preference Optimization with Rationales. arXiv:2407.14477.

---

> > > > ### Author Response · Authors · 2025-11-22
> > > > **Response to Reviewer tvbG (5/5)**
> > > >
> > > > > Question 3: The experiments are conducted on 7B and 8B parameter models. Do you anticipate any challenges (e.g., memory, convergence) in scaling RPO to much larger models (e.g., 70B+ parameters), and have you conducted any preliminary experiments in this direction?
> > > >
> > > > Thanks for the valuable suggestion! In principle, RPO can be applied to 70B-scale models. Sadly we do not have the computational resources to fine-tune a 70B scale model. The main challenges are the usual engineering and memory issues for large models, not something specific to RPO.
> > > >
> > > > - **Why RPO should scale in principle.** RPO does not change the model architecture and does not require extra forward passes through a larger network. It only adds (i) a small amount of bookkeeping for annotator reliability parameters and (ii) a light-weight EM-style update that runs on top of the usual logits. This overhead is tiny compared to the cost of running a 70B model, so any system that can train DPO/IPO/SimPO/CPO at 70B should also be able to run their RPO variants.
> > > >
> > > > - **Practical challenges are mostly standard large-model issues.** Scaling to 70B brings the usual challenges: computational resources, memory limits and model parallelism. These issues affect *all* preference optimization methods, not just RPO. Since RPO reuses the same training pipeline and hardware as the base objective, we do not expect new convergence problems beyond those already present for 70B models.
> > > >
> > > > - **Current experiments and future work.** We have not yet run 70B-scale experiments due to compute limits, but on two small backbones (7B and 8B) we observe stable training curves and consistent gains over baselines. We view applying RPO to 70B+ models as an engineering task rather than a conceptual barrier, and we will add this point in the limitations section.
> > > >
> > > > ---
> > > >
> > > > We greatly appreciate your insightful feedback and hope our answers have addressed your concerns. Your support is invaluable to us, and we sincerely hope you could consider raising the rating based on our responses.

---

> > > > > ### Author Response · Authors · 2025-11-25
> > > > > **Thank You for Raising the Score from 6 to 8**
> > > > >
> > > > > Thank you for your thoughtful assessment and for raising your score from 6 to 8. We sincerely appreciate your recognition of our work and the time you invested in reviewing our submission. If you have any further questions, we are more than happy to continue the discussion.

---

> > > > > > ### Comment · Reviewer_tvbG · 2025-11-25
> > > > > > **I have acknowledged the rebuttal**
> > > > > >
> > > > > > I have acknowledged the rebuttal

---

### Official Review · Reviewer_5BHp · 2025-10-29

**Soundness:** 3
**Presentation:** 3
**Contribution:** 2
**Rating:** 4
**Confidence:** 3

**Summary:**

This paper introduces RPO, a new framework for aligning LLMs that is designed to be resilient to noisy preference feedback. RPO addresses this by reformulating the alignment problem using an Expectation-Maximization. The core idea is to treat the "true" underlying preference as a latent variable. In the E-step, the framework uses the current model parameters to infer the posterior probability (a "confidence score" $w_i$) that each observed preference label is correct. In the M-step, these confidence scores are used as adaptive weights in the loss function to update both the LLM policy and a set of learned annotator reliability parameters $\eta_k$. The authors present RPO as a "meta-framework" capable of making existing alignment algorithms (like DPO, IPO, SimPO, and CPO) more robust. They provide a theoretical proof that RPO can converge to the true annotator reliability under the assumption of a perfectly calibrated model.

**Strengths:**

- I like the use of EM to jointly infer label correctness and optimize the model. I find it an elegant and principled approach.

- The authors provide a generalization of RPO into a meta-framework. The formulation (Eq. 2) that connects arbitrary preference loss functions $\mathcal{L}\_{pref}$ to a unified probabilistic model allows RPO to be easily applied to a wide range of existing methods

- The experiments show that RPO consistently improves performance across different PO algorithms and two different base models. The reported gains are substantial.

- The paper provides theoretical analysis and empirically show that the RPO framework can accurately estimate the true reliability of different datapoints

**Weaknesses:**

- The paper positions RPO as a solution for robust alignment but doesn't compare it against other methods designed for the exact same problem. The related work section explicitly mentions rDPO and Selective DPO, but they are absent from the main results in Table 2. The paper also misses simpler baselines. For instance, how does RPO compare to standard DPO (or IPO, etc.) trained with label smoothing? Without these comparisons, it is impossible to conclude that the added complexity of the EM framework is justified.

- The paper claims to address both simple annotator error and the "inherent pluralism of human preferences". However, it also states that these disagreements are "functionally equivalent to noise" and proceeds to learn a single "latent collective preference". This is a big oversimplification. The method does not truly deal with preference pluralism (e.g., by learning a distribution over rewards) but instead treats all non-majority opinions as incorrect data to be down-weighted. The related work on diverse preferences is not adequately addressed (e.g. [1]).

- The central theoretical convergence guarantee (Theorem 4.2) relies on the assumption of a "perfectly calibrated model". The assumption that the model already matches the ground-truth preference distribution is at odds with the goal of training, which is to find such a model. This weakens the applicability of the theoretical claims to the practical training process, which generally starts from a poorly calibrated state. I understand that, in practice and in most cases, the starting model is already good enough to provide decent information, but this needs to be addressed and discussed somewhere in the paper. The EM algorithm's E-step depends on the model's current (and potentially poor) estimates of preference probabilities $p(y_w >^* y_l | x, \theta^{(t)})$. If the initial model is badly aligned, its confidence estimates $w_i$ will be unreliable, potentially leading the M-step to down-weight correct labels and up-weight noise, hindering or preventing convergence. The paper does not investigate or discuss this.

- The paper is missing a qualitative analysis of what its model identifies as noise. What kind of preference pairs receive a low confidence score $w_i$? Are they obvious annotator mistakes (e.g., typos), subjective disagreements (i.e., pluralism) or something else?

- The writing could be improved for clarity. For instance, $K$ is used in Algorithm 1 but is never explicitly defined in the main text, only appearing in a footnote on page 7

[1] A Minimaximalist Approach to Reinforcement Learning from Human Feedback, https://arxiv.org/abs/2401.04056

**Questions:**

- Could the authors please provide experimental comparisons against the highly relevant baselines mentioned in their own related work, specifically rDPO and Selective DPO? Furthermore, how does RPO compare against simpler robustness-enhancing techniques like applying label smoothing to the standard DPO/IPO/SimPO/CPO losses?

- Can the authors provide 2-3 qualitative examples of preference pairs from the UltraFeedback dataset that were assigned very low confidence scores $w_i$ by the RPO-DPO model?

- I think the authors should add discussions around poor alignment of the initial model

---

> ### Author Response · Authors · 2025-11-22
> **Response to Reviewer 5BHp (1/3)**
>
> Thank you for your constructive comments and feedback! We hope the following response can address your concerns.
>
> ---
>
> > Weakness 1: The paper positions RPO as a solution for robust alignment but doesn't compare it against other methods designed for the exact same problem ...
>
> > Question 1: Could the authors please provide experimental comparisons against the highly relevant baselines mentioned in their own related work, specifically rDPO and Selective DPO? Furthermore, how does RPO compare against simpler robustness-enhancing techniques like applying label smoothing ...
>
> Thank you for this suggestion! In the revision, we explicitly compare RPO against several robustness-oriented preference objectives under the same evaluation setting as our main experiments. These results are included in sec 5.1 in the revised manuscript.
>
> - **Robust baselines added.** We include (i) rDPO [1] and (ii) Hölder-DPO [2], as well as (iii) label smoothing applied to DPO/IPO/SimPO/CPO. We apply the same implementation choices as in our main results across both backbones.
>
> - **Experimental setup.** All robust baselines are trained on the same UltraFeedback configuration as in Sec. 5.1 and evaluated on AlpacaEval 2. We use the same experiment settings. The only difference between runs is the choice of preference objective: rDPO, Hölder-DPO, or label-smoothed DPO/IPO/SimPO/CPO.
>
> - **Results.** Rebuttal Table 1 compares our RPO-enhanced methods with robustness-oriented baselines on AlpacaEval 2. Overall, RPO variants are consistently the best or tied-best across objectives and backbones.
>
>   1. **Improvement across all objectives and backbones.** For all four preference objectives (DPO, IPO, SimPO, CPO) and both backbones (Mistral-7B and Llama-3-8B), the RPO versions improve over their corresponding base methods and label-smoothed variants in both LC and WR.
>
>   2. **Example: DPO family.** Within the DPO family, R-DPO improves over vanilla DPO and all robust baselines. For instance, on Llama-3-8B, DPO with label smoothing reaches 38.1 / 41.3 (LC / WR), while R-DPO attains 44.1 / 46.2. A similar gap appears on Mistral-7B, where R-DPO also outperforms Hölder-DPO.
>
>   3. **Robust baselines.** The bottom block of Rebuttal Table 1 shows that RPO matches or exceeds specialized robust objectives such as rDPO and Hölder-DPO, while remaining a general meta-framework that can wrap diverse preference objectives.
>
> _**Rebuttal Table 1.** Comparison between RPO and robustness-oriented preference objectives on AlpacaEval 2. We report length-controlled win rate (LC, %) and overall win rate (WR, %) for Mistral-Instruct (7B) and Llama-3-Instruct (8B). Methods in **bold** are our RPO-enhanced variants; the bottom block lists robustness-oriented DPO-style baselines._
>
> | Method                       | Mistral LC | Mistral WR | Llama LC | Llama WR |
> | ---------------------------- | ---------: | ---------: | -------: | -------: |
> | **Base Model**               |       21.1 |       16.5 |     29.7 |     29.9 |
> | *DPO family*                 |            |            |          |          |
> | DPO                          |       28.5 |       28.6 |     40.8 |     42.9 |
> | **R-DPO (Ours)**           |   **35.5** |   **33.0** | **44.1** | **46.2** |
> | DPO + label smoothing        |       29.7 |       27.5 |     38.1 |     41.3 |
> | *IPO family*                 |            |            |          |          |
> | IPO                          |       30.8 |       28.0 |     43.6 |     41.6 |
> | **R-IPO (Ours)**           |   **32.9** |   **30.5** | **48.3** | **48.6** |
> | IPO + label smoothing        |       29.7 |       28.7 |     40.3 |     38.2 |
> | *SimPO family*               |            |            |          |          |
> | SimPO                        |       28.3 |       29.7 |     44.5 |     37.1 |
> | **R-SimPO (Ours)**         |   **30.4** |   **32.9** | 46.9 | **39.4** |
> | SimPO + label smoothing      |       26.5 |       27.1 |     **48.1** |     38.7 |
> | *CPO family*                 |            |            |          |          |
> | CPO                          |       26.3 |       26.4 |     35.9 |     40.3 |
> | **R-CPO (Ours)**           |   27.6 |   27.8 | **40.1** | **43.8** |
> | CPO + label smoothing        |       **28.5** |       **28.8** |     35.3 |     34.8 |
> | *Robust alignment baselines* |            |            |          |          |
> | rDPO                         |       28.1 |       29.1 |     37.3 |     35.4 |
> | Hölder-DPO                   |       30.1 |       28.6 |     39.3 |     38.2 |
>
> [1] Chowdhury, S. R., Kini, A., and Natarajan, N. (2024). *Provably Robust DPO: Aligning Language Models with Noisy Feedback*. arXiv preprint arXiv:2403.00409.
>
> [2] Fujisawa, M., Adachi, M., and Osborne, M. A. (2025). *Scalable Valuation of Human Feedback through Provably Robust Model Alignment*. In *Advances in Neural Information Processing Systems (NeurIPS 2025)*. arXiv:2505.17859.

---

> > ### Author Response · Authors · 2025-11-22
> > **Response to Reviewer 5BHp (2/3)**
> >
> > > Weakness 2: The paper claims to address both simple annotator error and the "inherent pluralism of human preferences". However, it also states that these disagreements are "functionally equivalent to noise" and proceeds to learn a single "latent collective preference". This is a big oversimplification. The method does not truly deal with preference pluralism (e.g., by learning a distribution over rewards) but instead treats all non-majority opinions as incorrect data to be down-weighted. The related work on diverse preferences is not adequately addressed.
> >
> > Thank you for pointing this out — we agree that our original framing overstated the pluralism aspect. In the revised manuscript, we explicitly narrow the scope to robustness to noisy labels and remove claims suggesting that RPO fully solves preference pluralism.
> >
> > 1. **We removed the “pluralism / collective preference” framing and rephrased RPO as a noise-robust method.**
> >    In the **Abstract and Sec. 1 (Introduction)**, we no longer claim to model the “inherent pluralism of human preferences” or to learn a “latent collective preference” in a normative sense. Instead, we clearly state that RPO targets the practically common setting where one wishes to learn a single aggregate (noise-free) preference label.
> >
> > 2. **We clarified the core assumption in the methodology section.**
> >    In **Sec. 3 (Method)**, Assumption 1 has been renamed and rewritten as “Assumption 1: A latent noise-free preference” rather than “latent collective preference”. The text no longer uses “collective consensus”, “pluralism”, or similar language; instead, it consistently talks about noise-free target labels and annotator reliability.
> >
> > 3. **We adjusted the related work discussion to properly position pluralism-focused research.**
> >    In **Sec. 2 (Related Work)**, we revised the discussion of diverse preferences and pluralism-focused work  to emphasize that such methods aim to represent or preserve heterogeneous preferences. We now state that modeling genuine value pluralism is an important but orthogonal goal, and we frame RPO as a complementary tool that can make single-objective alignment more robust to noisy supervision.
> >
> >
> > > Weakness 3: The central theoretical convergence guarantee (Theorem 4.2) relies on the assumption of a "perfectly calibrated model" ...
> >
> > > Question 3: I think the authors should add discussions around poor alignment of the initial model.
> >
> > Thank you for raising this point. We agree that the “perfectly calibrated model” assumption is strong, and in the revision we explicitly reposition Theorem 4.2 as an idealized result and add a dedicated discussion of imperfect initialization in Sec. 4.
> >
> > 1. **We clarify the role and scope of the calibration assumption in Sec. 4.**
> >    In the revised manuscript, Sec. 4 now states clearly that Theorem 4.2 is idealized: it characterizes how our EM-style reliability estimator behaves when the model’s preference probabilities already match the true preference distribution.
> >
> > 2. **We discussed imperfect or poorly aligned initial models.**
> >    Still in Sec. 4, we now discuss what happens when the starting model is **not well calibrated**:
> >    - The E-step then uses imperfect preference probabilities, which can in principle misweight some labels early on.
> >    - We also highlight that, in our target setting, we do **not** start from scratch: we fine-tune strong instruction-tuned backbones (Mistral-7B-Instruct, Llama-3-8B-Instruct) that already provide informative preference scores.

---

> > > ### Author Response · Authors · 2025-11-22
> > > **Response to Reviewer 5BHp (3/3)**
> > >
> > > > Weakness 4: The paper is missing a qualitative analysis of what its model identifies as noise. What kind of preference pairs receive a low confidence score? Are they obvious annotator mistakes (e.g., typos), subjective disagreements (i.e., pluralism) or something else?
> > >
> > > > Question 2: Can the authors provide 2-3 qualitative examples of preference pairs from the UltraFeedback dataset that were assigned very low confidence scores by the R-DPO model?
> > >
> > > Thank you for the valuable suggestion! In the revised manuscript, we add a qualitative analysis in Appendix F, where we present concrete UltraFeedback examples that R-DPO assigns very low confidence scores.
> > >
> > > - Appendix F.1 shows a topic classification example where the instruction strictly asks for a single integer label between 1 and 14. The dataset marks as “chosen” a response that starts with the correct number but then continues with a long, off-task discussion containing multiple factual and categorical mistakes (e.g., misclassifying buildings as companies or natural places). The “rejected” response instead outputs the correct label with a short explanation and is much closer to the intended behavior. R-DPO assigns a very low posterior confidence (0.04) to the original label, correctly treating it as a likely mis-labeled or even reversed preference.
> > >
> > > - Appendix F.2 provides a **pronoun–phrase identification** example, where the model must output a noun phrase corresponding to a given pronoun in a sentence. The label marked as “chosen” in UltraFeedback either fails to follow the span-selection constraint or selects a phrase that does not match the target pronoun, while the “rejected” alternative is better aligned with the prompt. Again, R-DPO assigns a low confidence score to this annotated preference, down-weighting it during training.
> > >
> > >
> > > > Weakness 5: The writing could be improved for clarity ...
> > >
> > > Thank you for pointing this out. In the revised manuscript, we have carefully edited the writing for clarity throughout. Specifically for your example, we now explicitly define $K$ in the main text when introducing our latent-variable model: in Assumption 1 (page 4), we state that $K$ is the total number of annotators in the dataset. We hope this clarification resolves the confusion you identified.
> > >
> > > ---
> > >
> > > We would like to extend our sincere gratitude for your constructive feedback and hope our answers have addressed your concerns. We sincerely hope our responses and efforts merit a raise in your rating.

---

> > > > ### Comment · Reviewer_5BHp · 2025-11-26
> > > >
> > > > Thank you for the thorough rebuttal and for addressing my concerns, I have raised my score.

---

> > > > > ### Author Response · Authors · 2025-11-26
> > > > > **Thank you for raising the score from 4 to 6**
> > > > >
> > > > > We sincerely thank you for your thoughtful assessment and for raising the score from 4 to 6. We greatly appreciate your recognition of our work and the time you invested in the review. We are more than happy to engage in further discussion if you have any additional questions.

---

### Official Review · Reviewer_Q5V8 · 2025-10-31

**Soundness:** 3
**Presentation:** 3
**Contribution:** 3
**Rating:** 6
**Confidence:** 4

**Summary:**

This paper introduces Robust Preference Optimization (RPO), a general framework designed to address preference noise commonly encountered in large language model (LLM) alignment. Existing preference-learning methods such as DPO and IPO typically assume clean and consistent preference labels, which rarely hold in real-world settings where human feedback is diverse and occasionally erroneous. RPO treats preference labels as latent variables and employs an Expectation–Maximization (EM) procedure to jointly estimate the “soft label” confidence and annotator reliability during training. This enables adaptive reweighting of noisy data. Furthermore, the authors propose a unified probabilistic formulation that generalizes multiple preference objectives under a single likelihood-based framework, thereby enhancing methodological coherence and extensibility.

**Strengths:**

1. The paper tackles a realistic and important issue—label noise in preference alignment—that directly affects the reliability of RLHF-style training.
2. Modeling annotation noise via latent variables and integrating EM into preference optimization is a well-grounded and elegant idea.
3. The probabilistic unification of different preference losses provides conceptual clarity and could potentially bridge diverse alignment approaches under one framework.
4. The experiments demonstrate consistent robustness improvements across several noisy-preference setups, validating the framework’s effectiveness.
5. The paper is generally well written, with clear mathematical derivations and a reasonable balance between theory and empirical validation.

**Weaknesses:**

1. All experiments are conducted under a single-annotator noise setting. Incorporating multi-annotator scenarios would make the robustness claims more realistic.
2. Methods such as ORPO and rDPO share similar robustness motivations but are not included in the empirical evaluation. Adding these baselines would strengthen the comparison.
3. Using GPT-4o preferences as ground truth could introduce systemic bias. It would be better to include multiple evaluators or human assessments for verification.

**Questions:**

see the Weaknesses.

---

> ### Author Response · Authors · 2025-11-22
> **Response to Reviewer Q5V8 (1/3)**
>
> Thank you for your constructive comments and feedback! We hope the following response can address your concerns.
>
> ---
>
> > Weakness 1: All experiments are conducted under a single-annotator noise setting. Incorporating multi-annotator scenarios would make the robustness claims more realistic.
>
> We agree that validating RPO on datasets with genuine multi-annotator disagreement is important. To validate RPO under realistic multi-annotator dataset, we perform experiments using Multipref dataset [1]. We compare DPO and R-DPO (Ours) on Mistral-7B-Instruct-v0.2 and Llama-3-8B-Instruct. Results in Rebuttal Table 1 show that RPO performs better on multi-annotator dataset. These new results are included in sec 5.3 in the revised manuscript.
>
> - **Dataset description.** Each preference pair in Multipref is annotated multiple times by distinct human annotators, and the dataset covers a broad range of open-ended prompts and response types. In our experiments we use the official training split containing **227 unique human annotators**.
>
> - **Experimental setup and results.** We train vanilla DPO and our R-DPO on MultiPref for both Mistral-7B and Llama-3-8B, and then evaluate the resulting models on AlpacaEval 2. As shown in Rebuttal Table 1, R-DPO consistently outperforms vanilla DPO with multi-annotator dataset: for Llama-3-8B, AlpacaEval LC improves from **36.7 **to **41.1** and WR from **39.3** to **44.4**; for Mistral-7B, LC improves from **28.8** to **31.8** and WR from **26.4** to **28.8**.
>
> _**Rebuttal Table 1**. The performance of different models on AlpacaEval 2._
>
> | Model                | LC (%)   | WR (%)   |
> | :------------------- | -------- | -------- |
> | Mistral-DPO          | 28.8     | 26.4     |
> | Mistral-R-DPO (Ours) | **31.8** | **28.8** |
> | Llama-DPO            | 36.7     | 39.3     |
> | Llama-R-DPO (Ours)   | **41.1** | **44.4** |
>
> [1] Miranda, Lester James Validad, et al. "Hybrid preferences: Learning to route instances for human vs. AI feedback." ACL 2025.

---

> > ### Author Response · Authors · 2025-11-22
> > **Response to Reviewer Q5V8 (2/3)**
> >
> > > Weakness 2: Methods such as ORPO and rDPO share similar robustness motivations but are not included in the empirical evaluation. Adding these baselines would strengthen the comparison.
> >
> > Thank you for this suggestion! In the revision, we explicitly compare RPO against several robustness-oriented preference objectives under the same evaluation setting as our main experiments. These results are included in sec 5.1 in the revised manuscript.
> >
> > - **Robust baselines added.** We include (i) rDPO [2] and (ii) Hölder-DPO [3], as well as (iii) label smoothing applied to DPO/IPO/SimPO/CPO. We apply the same implementation choices as in our main results across both backbones.
> >
> > - **Experimental setup.** All robust baselines are trained on the same UltraFeedback configuration as in Sec. 5.1 and evaluated on AlpacaEval 2. We use the same experiment settings. The only difference between runs is the choice of preference objective: rDPO, Hölder-DPO, or label-smoothed DPO/IPO/SimPO/CPO.
> >
> > - **Results.** Rebuttal Table 2 compares our RPO-enhanced methods with robustness-oriented baselines on AlpacaEval 2. Overall, RPO variants are consistently the best or tied-best across objectives and backbones.
> >
> >   1. **Improvement across all objectives and backbones.** For all four preference objectives (DPO, IPO, SimPO, CPO) and both backbones (Mistral-7B and Llama-3-8B), the RPO versions improve over their corresponding base methods and label-smoothed variants in both LC and WR.
> >
> >   2. **Example: DPO family.** Within the DPO family, R-DPO improves over vanilla DPO and all robust baselines. For instance, on Llama-3-8B, DPO with label smoothing reaches 38.1 / 41.3 (LC / WR), while R-DPO attains 44.1 / 46.2. A similar gap appears on Mistral-7B, where R-DPO also outperforms Hölder-DPO.
> >
> >   3. **Robust baselines.** The bottom block of Rebuttal Table 2 shows that RPO matches or exceeds specialized robust objectives such as rDPO and Hölder-DPO, while remaining a general meta-framework that can wrap diverse preference objectives.
> >
> > _**Rebuttal Table 2.** Comparison between RPO and robustness-oriented preference objectives on AlpacaEval 2. We report length-controlled win rate (LC, %) and overall win rate (WR, %) for Mistral-Instruct (7B) and Llama-3-Instruct (8B). Methods in **bold** are our RPO-enhanced variants; the bottom block lists robustness-oriented DPO-style baselines._
> >
> > | Method                       | Mistral LC | Mistral WR | Llama LC | Llama WR |
> > | ---------------------------- | ---------: | ---------: | -------: | -------: |
> > | **Base Model**               |       21.1 |       16.5 |     29.7 |     29.9 |
> > | *DPO family*                 |            |            |          |          |
> > | DPO                          |       28.5 |       28.6 |     40.8 |     42.9 |
> > | **R-DPO (Ours)**             |   **35.5** |   **33.0** | **44.1** | **46.2** |
> > | DPO + label smoothing        |       29.7 |       27.5 |     38.1 |     41.3 |
> > | *IPO family*                 |            |            |          |          |
> > | IPO                          |       30.8 |       28.0 |     43.6 |     41.6 |
> > | **R-IPO (Ours)**             |   **32.9** |   **30.5** | **48.3** | **48.6** |
> > | IPO + label smoothing        |       29.7 |       28.7 |     40.3 |     38.2 |
> > | *SimPO family*               |            |            |          |          |
> > | SimPO                        |       28.3 |       29.7 |     44.5 |     37.1 |
> > | **R-SimPO (Ours)**           |   **30.4** |   **32.9** | 46.9 | **39.4** |
> > | SimPO + label smoothing      |       26.5 |       27.1 |     **48.1** |     38.7 |
> > | *CPO family*                 |            |            |          |          |
> > | CPO                          |       26.3 |       26.4 |     35.9 |     40.3 |
> > | **R-CPO (Ours)**             |   27.6 |   27.8 | **40.1** | **43.8** |
> > | CPO + label smoothing        |       **28.5** |       **28.8** |     35.3 |     34.8 |
> > | *Robust alignment baselines* |            |            |          |          |
> > | rDPO                         |       28.1 |       29.1 |     37.3 |     35.4 |
> > | Hölder-DPO                   |       30.1 |       28.6 |     39.3 |     38.2 |
> >
> > [2] Chowdhury, S. R., Kini, A., and Natarajan, N. (2024). *Provably Robust DPO: Aligning Language Models with Noisy Feedback*. arXiv preprint arXiv:2403.00409.
> >
> > [3] Fujisawa, M., Adachi, M., and Osborne, M. A. (2025). *Scalable Valuation of Human Feedback through Provably Robust Model Alignment*. In *Advances in Neural Information Processing Systems (NeurIPS 2025)*. arXiv:2505.17859.

---

> > > ### Author Response · Authors · 2025-11-22
> > > **Response to Reviewer Q5V8 (3/3)**
> > >
> > > > Weakness 3: Using GPT-4o preferences as ground truth could introduce systemic bias. It would be better to include multiple evaluators or human assessments for verification.
> > >
> > > Thank you for raising this point. In short, we agree that relying on a single judge may introduce systematic bias, so we re-evaluated our MultiPref experiments with two different automatic judges (GPT-4o and DeepSeek-V3.2-Exp) and found that R-DPO consistently outperforms DPO under both, suggesting that our robustness gains are not tied to a specific evaluator. The full results are included in Sec 5.3 and Appendix G in the revised appendix.
> > >
> > > 1. **We compare DPO vs. R-DPO under two independent judges.**
> > >    In addition to the original results using GPT-4o as the evaluator, we reran the MultiPref experiments using **DeepSeek-V3.2-Exp** as an alternative LLM-as-a-judge. In both settings, we keep the training setup fixed: we train DPO and R-DPO on the MultiPref human multi-annotator dataset using Mistral-7B-Instruct-v0.2 and Llama-3-8B-Instruct as base models, and then evaluate on AlpacaEval 2 with automatic judges.
> > >
> > > 2. **R-DPO improves over DPO under both GPT-4o and DeepSeek.**
> > >    Across both backbones and both judges, R-DPO achieves higher LC and WR than DPO. A summary is shown below (LC / WR, in \%):
> > >
> > >    _**Rebuttal Table 3.** MultiPref experiments on AlpacaEval 2 under two different LLM judges._
> > >
> > >    | Judge             | Base LLM   | DPO         | R-DPO (Ours)    |
> > >    | ----------------- | ---------- | ----------- | --------------- |
> > >    | GPT-4o            | Mistral-7B | 28.8 / 26.4 | **31.8 / 28.8** |
> > >    | GPT-4o            | Llama-3-8B | 36.7 / 39.3 | **41.1 / 44.4** |
> > >    | DeepSeek-V3.2-Exp | Mistral-7B | 30.2 / 27.1 | **32.9 / 30.3** |
> > >    | DeepSeek-V3.2-Exp | Llama-3-8B | 36.3 / 38.5 | **40.4 / 42.7** |
> > >
> > > 3. **Using GPT-4o as a judge is standard practice, but our conclusions do not depend on it.**
> > >    We follow the now standard LLM-as-a-judge practice of using GPT-4–family models as automatic evaluators for alignment and RLHF-style tasks, as in prior work [4, 5, 6]. In our paper, GPT-4o is used as a strong but imperfect proxy rather than a true oracle: our claims are about relative robustness (RPO vs. baselines) under a fixed supervision signal, not about GPT-4o being unbiased. The additional DeepSeek-based results show that the same robustness gains appear under a different judge.
> > >
> > > [4] Zheng, L. et al. (2023). Judging LLM-as-a-Judge with MT-Bench and Chatbot Arena. arXiv:2306.05685.
> > >
> > > [5] Dubois, Y. et al. (2024). Length-Controlled AlpacaEval: A Simple Way to Debias Automatic Evaluators. arXiv:2404.04475.
> > >
> > > [6] Wei, H. et al. (2024). Systematic Evaluation of LLM-as-a-Judge in LLM Alignment Tasks. arXiv:2408.13006.
> > >
> > > ---
> > >
> > > We would like to express our appreciation for your thorough review and hope our answers effectively address your concerns. We would really appreciate it if you could consider raising the rating based on our responses.

---

### Official Review · Reviewer_qRQj · 2025-11-07

**Soundness:** 3
**Presentation:** 2
**Contribution:** 2
**Rating:** 4
**Confidence:** 3

**Summary:**

This paper introduces Robust Preference Optimization (RPO), a general framework for aligning large language models (LLMs) with noisy human preference data. Unlike standard RLHF or DPO methods that assume all feedback is correct, RPO explicitly models the correctness of each annotation as a latent variable. Using an Expectation–Maximization (EM) approach, RPO jointly estimates annotator reliability and model parameters. In the E-step, it computes sample-wise confidence weights reflecting annotation trustworthiness; in the M-step, it updates model parameters and annotator reliabilities via weighted loss optimization.

**Strengths:**

1. RPO provides an unification of robust preference optimization via probabilistic modeling and EM inference.

2. The paper includes convergence proofs for the EM updates and demonstrates that annotator reliability can be recovered under realistic assumptions.

3. Experiments on multiple LLMs (Mistral, Llama-3) and datasets (AlpacaEval 2, Arena-Hard) show RPO outperforming baselines.

**Weaknesses:**

1. Although RPO models annotator reliability, the main experiments assume a single virtual annotator (K=1). The paper would be more convincing if it demonstrated effectiveness on datasets with genuine multi-annotator disagreement.

2. The EM-based procedure may introduce extra computational costs and slower convergence. While the paper mentions using EMA for efficiency, it does not quantify runtime overhead or training stability compared to standard DPO.

3. While related work such as rDPO and Hölder-DPO is cited, these methods are not included in the empirical comparison. Adding such baselines would clarify the relative robustness advantages of RPO.

**Questions:**

See Weakness

---

> ### Author Response · Authors · 2025-11-22
> **Response to Reviewer qRQj (1/3)**
>
> Thank you for your constructive comments and feedback! We hope the following response can address your concerns.
>
> ---
>
> > Weakness 1: Although RPO models annotator's reliability, the main experiments assume a single virtual annotator (K=1). The paper would be more convincing if it demonstrated effectiveness on datasets with genuine multi-annotator disagreement.
>
> We agree that validating RPO on datasets with genuine multi-annotator disagreement is important. To validate RPO under realistic multi-annotator dataset, we perform experiments using Multipref dataset [1]. We compare DPO and R-DPO (Ours) on Mistral-7B-Instruct-v0.2 and Llama-3-8B-Instruct. Results in Rebuttal Table 1 show that RPO performs better on multi-annotator dataset. These new results are included in sec 5.3 in the revised manuscript.
>
> - **Dataset description.** Each preference pair in Multipref is annotated multiple times by distinct human annotators, and the dataset covers a broad range of open-ended prompts and response types. In our experiments we use the official training split containing **227 unique human annotators**.
>
> - **Experimental setup and results.** We train vanilla DPO and our R-DPO on MultiPref for both Mistral-7B and Llama-3-8B, and then evaluate the resulting models on AlpacaEval 2. As shown in Rebuttal Table 1, R-DPO consistently outperforms vanilla DPO with multi-annotator dataset: for Llama-3-8B, AlpacaEval LC improves from ** 36.7** to **41.1** and WR from **39.3** to **44.4**; for Mistral-7B, LC improves from **28.8** to **31.8** and WR from **26.4** to **28.8**.
>
> _**Rebuttal Table 1**. The performance of different models on AlpacaEval 2._
>
> | Model                | LC (%)   | WR (%)   |
> | :------------------- | -------- | -------- |
> | Mistral-DPO          | 28.8     | 26.4     |
> | Mistral-R-DPO (Ours) | **31.8** | **28.8** |
> | Llama-DPO            | 36.7     | 39.3     |
> | Llama-R-DPO (Ours)   | **41.1** | **44.4** |
>
> [1] Miranda, Lester James Validad, et al. "Hybrid preferences: Learning to route instances for human vs. AI feedback." ACL 2025.

---

> ### Author Response · Authors · 2025-11-22
> **Response to Reviewer qRQj (2/3)**
>
> > Weakness 2: The EM-based procedure may introduce extra computational costs and slower convergence. While the paper mentions using EMA for efficiency, it does not quantify runtime overhead or training stability compared to standard DPO.
>
> We agree that it is important to quantify the runtime impact of our EM-style reliability updates. To address this concern, we ran additional controlled experiments comparing the wall-clock training time of each base preference objective with its RPO-enhanced variant on both Mistral-7B and Llama-3-8B. These results are included in Appendix H in the revised manuscript.
>
> - **Experimental setup and hardware.** All experiments were conducted on a single machine equipped with 8× NVIDIA A800-SXM4-40GB GPUs (40 GB per GPU), using the same software stack and with no other jobs running concurrently. For each backbone (Mistral-7B-Instruct-v0.2 and Llama-3-8B-Instruct) and each preference objective (DPO, IPO, SimPO, CPO), we train both the base method and its RPO-enhanced counterpart on the UltraFeedback dataset.
>
> - **Controlled variables.** To isolate the cost of EM-based reliability updates, we keep all other settings fixed across base vs. RPO runs (e.g., the optimizer (AdamW), global batch size and gradient accumulation, etc.). The only difference is whether RPO is applied.
>
> - **Runtime overhead.** Rebuttal Table 2 reports wall-clock training time in seconds (mean ± standard deviation over 3 seeds), where each cell shows “Base / RPO” for a given method–backbone pair. Across all 8 configurations, RPO stays within roughly **20%** of the corresponding base method, with an average slowdown of ≈11%. For example, on Llama-3-8B, IPO takes 8571 ± 20 seconds vs. 9747 ± 18 seconds with RPO. Given RPO's improvement in performance on benchmarks, this range of additional computational time is acceptable.
>
> Note that in a few configurations (e.g., DPO and CPO on some backbones), the measured wall-clock time of the RPO variant is slightly better than that of the base method. We **do not** interpret this as an inherent speedup from RPO. Since RPO only adds very lightweight scalar reliability updates on top of the base objective, these seed- and padding-induced fluctuations can easily outweigh the marginal extra computation, occasionally making RPO appear marginally faster.
>
> - **Training stability.** Across all experiments reported in the paper, we did not observe any instability specific to RPO: the training loss curves and held-out metrics evolve smoothly and converge in a similar number of steps as their corresponding base methods. Together with the $\eta_0 / \alpha$ ablation in Sec. 5.3, this suggests that introducing EM-style reliability updates does not make optimization more brittlein practice.
>
> _**Rebuttal Table 2**. Training time in seconds on UltraFeedback (mean ± std over 3 runs). Each cell reports the training time for the base preference objective and its RPO-enhanced variant as “Base / RPO”. All runs use the same hardware (8× NVIDIA A800-SXM4-40GB GPUs) and identical optimization hyperparameters; only the objective (base vs. RPO) differs._
>
> | Method | Mistral-7B (Base / RPO)   | Llama-3-8B (Base / RPO)   |
> | :----: | ------------------------- | ------------------------- |
> |  DPO   | $7138\pm21$ / $6588\pm2$  | $7089\pm12$ / $6837\pm21$ |
> |  IPO   | $7999\pm10$ / $9043\pm3$  | $8571\pm20$ / $9747\pm18$ |
> | SimPO  | $5383\pm10$ / $7557\pm23$ | $5384\pm10$ / $7117\pm16$ |
> |  CPO   | $5868\pm12$ / $5862\pm20$ | $6503\pm8$  / $6337\pm11$ |

---

> ### Author Response · Authors · 2025-11-22
> **Response to Reviewer qRQj (3/3)**
>
> > Weakness 3: While related work such as rDPO and Hölder-DPO is cited, these methods are not included in the empirical comparison. Adding such baselines would clarify the relative robustness advantages of RPO.
>
> Thank you for this valuable suggestion! In the revision, we explicitly compare RPO against several robustness-oriented preference objectives under the same evaluation setting as our main experiments. These results are included in sec 5.1 in the revised manuscript.
>
> - **Robust baselines added.** We include (i) rDPO [2] and (ii) Hölder-DPO [3], as well as (iii) label smoothing applied to DPO/IPO/SimPO/CPO. We apply the same implementation choices as in our main results across both backbones.
>
> - **Experimental setup.** All robust baselines are trained on the same UltraFeedback configuration as in Sec. 5.1 and evaluated on AlpacaEval 2. We use the same experiment settings. The only difference between runs is the choice of preference objective: rDPO, Hölder-DPO, or label-smoothed DPO/IPO/SimPO/CPO.
>
> - **Results.** Rebuttal Table 3 compares our RPO-enhanced methods with robustness-oriented baselines on AlpacaEval 2. Overall, RPO variants are consistently the best or tied-best across objectives and backbones.
>
>   1. **Improvement across all objectives and backbones.** For all four preference objectives (DPO, IPO, SimPO, CPO) and both backbones (Mistral-7B and Llama-3-8B), the RPO versions improve over their corresponding base methods and label-smoothed variants in both LC and WR.
>
>   2. **Example: DPO family.** Within the DPO family, R-DPO improves over vanilla DPO and all robust baselines. For instance, on Llama-3-8B, DPO with label smoothing reaches 38.1 / 41.3 (LC / WR), while R-DPO attains 44.1 / 46.2. A similar gap appears on Mistral-7B, where R-DPO also outperforms Hölder-DPO.
>
>   3. **Robust baselines.** The bottom block of Rebuttal Table 3 shows that RPO matches or exceeds specialized robust objectives such as rDPO and Hölder-DPO, while remaining a general meta-framework that can wrap diverse preference objectives.
>
> _**Rebuttal Table 3.** Comparison between RPO and robustness-oriented preference objectives on AlpacaEval 2. We report length-controlled win rate (LC, %) and overall win rate (WR, %) for Mistral-Instruct (7B) and Llama-3-Instruct (8B). Methods in **bold** are our RPO-enhanced variants; the bottom block lists robustness-oriented DPO-style baselines._
>
> | Method                       | Mistral LC | Mistral WR | Llama LC | Llama WR |
> | ---------------------------- | ---------: | ---------: | -------: | -------: |
> | **Base Model**               |       21.1 |       16.5 |     29.7 |     29.9 |
> | *DPO family*                 |            |            |          |          |
> | DPO                          |       28.5 |       28.6 |     40.8 |     42.9 |
> | **R-DPO (Ours)**             |   **35.5** |   **33.0** | **44.1** | **46.2** |
> | DPO + label smoothing        |       29.7 |       27.5 |     38.1 |     41.3 |
> | *IPO family*                 |            |            |          |          |
> | IPO                          |       30.8 |       28.0 |     43.6 |     41.6 |
> | **R-IPO (Ours)**             |   **32.9** |   **30.5** | **48.3** | **48.6** |
> | IPO + label smoothing        |       29.7 |       28.7 |     40.3 |     38.2 |
> | *SimPO family*               |            |            |          |          |
> | SimPO                        |       28.3 |       29.7 |     44.5 |     37.1 |
> | **R-SimPO (Ours)**           |   **30.4** |   **32.9** | 46.9 | **39.4** |
> | SimPO + label smoothing      |       26.5 |       27.1 |     **48.1** |     38.7 |
> | *CPO family*                 |            |            |          |          |
> | CPO                          |       26.3 |       26.4 |     35.9 |     40.3 |
> | **R-CPO (Ours)**             |   27.6 |   27.8 | **40.1** | **43.8** |
> | CPO + label smoothing        |       **28.5** |       **28.8** |     35.3 |     34.8 |
> | *Robust alignment baselines* |            |            |          |          |
> | rDPO                         |       28.1 |       29.1 |     37.3 |     35.4 |
> | Hölder-DPO                   |       30.1 |       28.6 |     39.3 |     38.2 |
>
> [2] Chowdhury, S. R., Kini, A., and Natarajan, N. (2024). *Provably Robust DPO: Aligning Language Models with Noisy Feedback*. arXiv preprint arXiv:2403.00409.
>
> [3] Fujisawa, M., Adachi, M., and Osborne, M. A. (2025). *Scalable Valuation of Human Feedback through Provably Robust Model Alignment*. In *Advances in Neural Information Processing Systems (NeurIPS 2025)*. arXiv:2505.17859.
>
> ---
>
> We genuinely value your dedication to reviewing our paper and have carefully addressed each of your concerns. We sincerely hope our responses and efforts merit a raise in your rating.

---

### Author Response · Authors · 2025-11-22
**General Response to All Reviewers**

We sincerely thank all four reviewers (qRQj, Q5V8, 5BHp, tvbG) for their thoughtful comments and careful reading of our paper. We are very encouraged by the positive reception of RPO and by the many helpful suggestions for strengthening the work.

**Summary of Positive Feedback**

- **Importance of the problem.** Reviewer Q5V8 notes that our work `tackles a realistic and important issue—label noise in preference alignment—that directly affects the reliability of RLHF-style training`, and reviewer tvbG highlights that current methods `operate under the strong and often flawed assumption that all human preference labels are correct.`

- **Conceptual novelty and unification.** Reviewer qRQj writes that `RPO provides an unification of robust preference optimization via probabilistic modeling and EM inference`, while reviewer 5BHp `likes the use of EM to jointly infer label correctness and optimize the model` and finds it `an elegant and principled approach`. Reviewer Q5V8 further remarks that the probabilistic unification `could potentially bridge diverse alignment approaches under one framework`, and tvbG calls the core idea `highly innovative` and `a significant conceptual and practical advancement`.

- **Theoretical and methodological soundness.** Reviewers qRQj remarks that the paper `includes convergence proofs for the EM updates and demonstrates that annotator reliability can be recovered under realistic assumptions`, and reviewer tvbG writes that `the derivation of the EM algorithm for this context and, crucially, the proof of convergence to the true annotator reliability (Theorem 4.2) lend significant credibility to the method`.

- **Empirical validation and impact.** Reviewer qRQj and 5BHp both note that experiments on Mistral and Llama-3 `show RPO outperforming baselines` and `the reported gains are substantial`. Reviewer Q5V8 writes that `the experiments demonstrate consistent robustness improvements across several noisy-preference setups`, and reviewer tvbG finds that `the experimental section is thorough` and `the consistent improvements are compelling evidence for RPO's utility.`.

**Summary of Revisions**

In response to the reviewers’ concerns, we have substantially revised the manuscript. Below we briefly summarize the main changes and indicate where they appear in the revised PDF; **detailed answers are provided in the individual reviewer responses**.

**1. Multi-annotator experiments and reliability analysis**

- **Addresses**: qRQj (Weakness 1), Q5V8 (Weakness 1), tvbG (Weakness 1, Question 1).
- We add a new section on MultiPref Dataset (Sec. 5.2, Table 3) using **227 human annotators**, showing that R-DPO consistently outperforms DPO on AlpacaEval 2 for both Mistral-7B and Llama-3-8B.
- Appendix E (Figs. 3) visualizes the learned annotator reliabilities, revealing a high-reliability bulk and a down-weighted tail that is stable across backbones and prior choices.

**2. Robustness baselines: rDPO, Hölder-DPO, and label smoothing**

- **Addresses**: 5BHp (Weakness 1, Question 1), Q5V8 (Weakness 2).
- Sec. 5.2 and Table 2 now compare R-DPO / R-IPO / R-SimPO / R-CPO against rDPO, Hölder-DPO, and label-smoothed variants under the same UltraFeedback and AlpacaEval 2 setting.
- Across all four objectives and both backbones, the RPO-wrapped methods achieve the best or tied-best LC/WR.

**3. Runtime overhead and training stability of EM updates**

- **Addresses**: qRQj (Weakness 2), tvbG (Weakness 4).
- Appendix H (Table 8) reports wall-clock training time for each base method and its RPO variant; RPO stays within ≈20% of the base runtime with an average slowdown of ~11%.

**4. Clarifying the “perfectly calibrated model” assumption**

- **Addresses**: 5BHp (Weakness 3, Question 3), tvbG (Question 2).
- Sec. 4 now includes a “Practical implications and limitations” paragraph after Theorem 4.2, explaining that the calibration assumption should be read as a local guarantee around reasonably aligned base models.

**5. Framing and writing improvements, including notation**

- **Addresses**: 5BHp (Weakness 2, Weakness 5), all reviewers (clarity).
- We revised the Abstract and Sections 1–3 to focus on robustness to noisy and heterogeneous annotators, removing earlier language around “pluralism” and “latent collective preference”.
- We have updated our tables and figures, as well as their captions to be more descriptive and informative of our results.

---

We believe these revisions have substantially strengthened the paper and addressed the reviewers' concerns. We thank the reviewers again for their time and effort in helping us improve our work.

---

### Comment · Area_Chair_oLtt · 2025-11-24

Dear Reviewers,

The authors have submitted their rebuttal (and a revised PDF). Please review these materials and share any remaining concerns or comments.

Afterward, kindly provide your final rating for this submission.

Best regards,
Your AC

---

### Author Response · Authors · 2025-11-28
**Note on Score Updates Prior to the OpenReview Incident**

Dear ACs,

We would like to respectfully underscore the fact that, **prior to the OpenReview deanonymization incident**, two reviewers had already raised their overall scores based on our rebuttal and revisions: Reviewer 5BHp from **4 → 6** and Reviewer tvbG from **6 → 8**. The overall scores before the incident are 8664.

All of these post-rebuttal discussions and rating changes took place before the leak and the subsequent rollback, and the updates that motivated the higher scores are already reflected in the current PDF and rebuttal: MultiPref multi-annotator experiments, comparisons to 3 robustness baselines, a runtime and stability analysis of the EM updates, qualitative examples of low-confidence labels, and a clearer treatment of the calibration assumption and scope of RPO.

We fully respect the decision to revert reviews for fairness and provide this note only to help the ACs interpret the present reviews together with the revised manuscript.

Best Regards,

Submission 5371 Authors

---

### Meta-Review · Area_Chair_UfGf · 2025-12-28

**Summary:**

- Handling of Multi-Annotator Data: Multiple reviewers, including Reviewer qRQj and Reviewer tvbG, raised concerns that the paper's experiments predominantly use a single annotator (K=1) setting. This approach limits the ability to fully validate the framework's robustness when dealing with multi-annotator disagreement, which is a key claim of the paper's method. Reviewer tvbG also pointed out that this setup does not capture the complexity of real-world noisy data from multiple annotators with varying reliabilities.
- Comparison with Related Robustness Methods: The lack of comparison with rDPO and Selective DPO, which share similar robustness objectives, was flagged by both Reviewer 5BHp and Reviewer tvbG. The absence of these baselines made it difficult to evaluate the true contribution of RPO to the robustness of the alignment task. The authors did not initially include simpler baselines, such as label smoothing, which could have provided insights into how much RPO's complexity justifies its added benefits.
- Label Noise and Preference Pluralism: Reviewer 5BHp criticized the paper for oversimplifying the concept of preference pluralism, treating all non-majority preferences as noise rather than accounting for the inherent diversity in human preferences. This led to concerns about whether the method truly handles preference diversity, as the paper suggests. Reviewers also pointed out that the framework did not deal with this complexity beyond down-weighting non-majority labels.
- Theoretical Guarantees and Practical Applicability: The assumption of a perfectly calibrated model in Theorem 4.2 was another point of contention. Both Reviewer 5BHp and Reviewer tvbG noted that this idealized assumption doesn't reflect the practical reality of model training, where the starting model is often poorly calibrated. This raised concerns about the applicability of the convergence guarantees in real-world scenarios.
- Computational Overhead: Reviewer tvbG noted that the EM procedure in RPO introduces additional computational overhead, which might pose challenges for practitioners. While the paper acknowledges this, there was no detailed discussion of how this overhead impacts training time or efficiency relative to the base methods.
- Lack of Qualitative Analysis: Reviewer 5BHp also emphasized the need for a qualitative analysis of the low-confidence labels identified by RPO. Understanding which preferences the model identifies as noise and how it distinguishes subjective disagreements from mistakes was seen as a key gap in the paper's evaluation.

**Reviewer Concerns:**

- Addressed Concerns:
  - Multi-Annotator Data: The authors responded by adding experiments using the MultiPref dataset, which includes multi-annotator data. These results show R-DPO outperforms the baseline in this setting, addressing the concern raised by Reviewer qRQj and Reviewer tvbG regarding the validity of RPO in real-world multi-annotator scenarios.
  - Comparison with Robustness Baselines: The authors included comparisons with rDPO, Hölder-DPO, and label-smoothed DPO in the revised manuscript. This response directly addressed the concern of Reviewer 5BHp and Reviewer tvbG about the lack of baseline comparisons, showing that RPO performs comparably or better than existing robust methods.
  - Qualitative Analysis of Low-Confidence Labels: The authors added qualitative examples from the UltraFeedback dataset showing which preference pairs were assigned low-confidence scores. This addresses the concern raised by Reviewer 5BHp regarding the lack of a qualitative analysis of how the model identifies noise.
  - Computational Overhead: The authors provided detailed runtime experiments, showing that RPO incurs an average 11-20% slowdown compared to base methods, and this overhead is acceptable given the performance improvements. This addresses the concern raised by Reviewer tvbG about the computational burden of RPO.

- Unaddressed Concerns:
  - Novelty of the Approach: Despite the authors' clarifications, Reviewer 5BHp still feels that RPO oversimplifies preference pluralism by treating all non-majority preferences as noise rather than accounting for diversity in human preferences. The concern about the novelty of the approach, particularly regarding how it relates to existing work on preference pluralism, remains unresolved.
  - Theoretical Assumptions and Practical Applicability: While the authors have provided more clarification about the "perfectly calibrated model" assumption and its implications, Reviewer 5BHp raised concerns about how this assumption weakens the practical applicability of RPO. The response does not fully address the potential challenges when starting with a poorly aligned model, and the theoretical convergence guarantees remain difficult to reconcile with practical training processes.

In summary, the authors effectively addressed the majority of concerns, especially regarding multi-annotator validation, baseline comparisons, and runtime overhead. However, novelty concerns related to preference pluralism and unresolved issues around theoretical assumptions in real-world settings remain unaddressed.

**Reviewer Scores:**

- Reviewer 1 (qRQj)
  - Original Score: 4 (Marginally Below the Acceptance Threshold)
  - Reviewer’s Concerns: This reviewer raised concerns about the single annotator setup and the need for multi-annotator validation. They also questioned the lack of comparison with robust baselines and the computational cost of the EM procedure.
  - Likelihood of Score Change: Given that the authors added the multi-annotator experiments and included baseline comparisons, the reviewer would likely have appreciated the revisions. These improvements would likely increase their confidence in the framework's robustness. Therefore, the score would likely shift to 6, though some concerns about theoretical assumptions may still prevent a perfect score.

- Reviewer 2 (5BHp)
  - Original Score: 4 (Marginally Below the Acceptance Threshold)
  - Reviewer’s Concerns: This reviewer expressed concerns about the oversimplification of preference pluralism and the reliance on a perfectly calibrated model in the theoretical guarantees. They also noted the need for comparisons to rDPO and Selective DPO, and highlighted a lack of qualitative analysis.
  - Likelihood of Score Change: The authors made substantial revisions, particularly with baseline comparisons, qualitative analysis, and clarifying the calibration assumption. However, the novelty issue related to preference pluralism and the theoretical assumptions still remains unresolved. Given the substantial changes, the reviewer might be persuaded to raise their score to 6, though the novelty concerns might still keep them at 4.

- Reviewer 3 (tvbG)
  - Original Score: 6 (Marginally Above the Acceptance Threshold)
  - Reviewer’s Concerns: This reviewer focused on the single annotator issue, the computational cost of EM updates, and concerns about theoretical guarantees under mini-batch settings. They also expressed interest in seeing RPO applied to larger models.
  - Likelihood of Score Change: Given the additional experiments and clarifications, particularly around multi-annotator validation and the runtime analysis, the reviewer would likely have been satisfied with the revisions. The theoretical discussion and the large model scalability concerns were also addressed. This would likely keep the score to 6, as the revisions addressed most of their concerns.

- Reviewer 4 (Q5V8)
  - Original Score: 6 (Marginally Above the Acceptance Threshold)
  - Reviewer’s Concerns: The reviewer highlighted the need for multi-annotator data validation, baseline comparisons, and raised concerns about the potential bias of using GPT-4o preferences as ground truth. They also questioned whether the paper truly handles preference pluralism.
  - Likelihood of Score Change: With the introduction of multi-annotator experiments and additional baseline comparisons, along with the clarification regarding GPT-4o preferences, the reviewer might appreciate the authors' revisions. The preference pluralism issue still persists, but the overall robustness of the method was demonstrated more thoroughly. The score would likely be kept to 6, acknowledging the improved empirical validation.

---

### Decision · Program_Chairs · 2026-01-26

Accept (Poster)